

# Predicting soil carbon by efficiently using variation in a mid-IR soil spectral library

Anatol Helfenstein[1,2], Philipp Baumann[1], Raphael Viscarra Rossel[3], Andreas Gubler[4], Stefan Oechslin[5], and Johan Six[1]

[1]Department of Environmental Systems Science, Swiss Federal Institute of Technology, ETH-Zurich, Universitätsstrasse 2, 8092 Zürich, Switzerland
[2]Soil Geography and Landscape Group, Wageningen University, PO Box 47, 6700 AA Wageningen, The Netherlands
[3]School of Molecular and Life Sciences, Faculty of Science and Engineering, Curtin University, Perth, Western Australia, Australia
[4]Swiss Soil Monitoring Network (NABO), Agroscope, Reckenholzstrasse 191, 8046 Zürich, Switzerland
[5]School of Agricultural, Forest and Food Sciences HAFL, Bern University of Applied Sciences BFH, Bern, Switzerland

**Correspondence:** Anatol Helfenstein, Soil Geography and Landscape Group, Wageningen University, PO Box 47, 6700 AA Wageningen, The Netherlands (anatol.helfenstein@wur.nl)

**Abstract.** Traditional laboratory methods of acquiring soil information remain important for assessing key soil properties, soil functions and ecosystem services over space and time. Infrared spectroscopic modelling can link and massively scale up these methods for many soil characteristics in a cost-effective and timely manner. In Switzerland, only $10\,\%$ to $15\,\%$ of agricultural soils have been mapped sufficiently to serve spatial decision support systems, presenting an urgent need for rapid

quantitative soil characterization. The current Swiss soil spectral library (SSL; $n = 4374$) in the mid-infrared range includes soil samples from the Biodiversity Monitoring Program (BDM), arranged in a regularly spaced grid across Switzerland, and temporally-resolved data from the Swiss Soil Monitoring Network (NABO). Given the relatively low representation of organic soils and their organo-mineral diversity in the SSL, we aimed to develop both an efficient calibration sampling scheme and accurate modelling strategy to estimate soil carbon (SC) contents of heterogeneous samples between $0\,\mathrm{m}$ to $2\,\mathrm{m}$ depth from

26 locations within two drained peatland regions (HAFL dataset; $n = 116$). The focus was on minimizing the need for new reference analyses by efficiently mining the spectral information of SSL instances and their target-feature representations.

We used partial least square regressions (PLSR) together with a 5 times repeated, grouped by location, 10-fold cross validation (CV) to predict SC ranging from $1\,\%$ to $52\,\%$ in the local HAFL dataset. We compared the validation performance of different calibration schemes involving local models (1), models using the entire SSL spiked with local samples (2) and

subsets of local and SSL samples using the RS-LOCAL algorithm (3). Using local and RS-LOCAL calibrations with at least 5 local samples, we achieved similar validation results for predictions of SC up to $52\,\%$ ($R^2 = 0.94 - 0.96$, bias $= -0.6 - 1.5$, RMSE $= 2.6\,\%$ to $3.5\,\%$ total carbon). However, calibrations of representative SSL and local samples using RS-LOCAL only required 5 local samples for very accurate models (RMSE $= 2.9\,\%$ total carbon), while local calibrations required 50 samples for similarly accurate results (RMSE $< 3\,\%$ total carbon). Of the three approaches, the entire SSL spiked with local samples for

model calibration led to validations with the lowest performance in terms of $R^2$, bias and RMSE. Hence, we show that a simple and comprehensible modelling approach using RS-LOCAL together with a SSL is an efficient and accurate strategy when using





infrared spectroscopy. It decreases field and laboratory work, the bias of SSL-spiking approaches and the uncertainty of local models. If adequately mined, the information in a SSL is sufficient to predict SC in new and independent study regions, even if the local soil characteristics are very different from the ones in the SSL. This will help to efficiently scale up the acquisition of quantitative soil information over space and time.

**Keywords.** soil spectroscopy, soil spectral library, soil carbon, peat soil, rs-local, PLSR, transfer learning, spiking

# 1 Introduction

Soil, the "skin" of the earth, is a vital part of the natural environment and essential for global ecosystem services, including food and fiber production, water filtration, climate regulation and carbon sequestration (Schmidt et al., 2011; Tiessen et al., 1994). We gained our scientific understanding of soil through long and strenuous soil surveys complemented by careful chemical, physical, mineralogical, and biological laboratory analysis. These conventional methodologies continue to be important for understanding complex soil processes, especially at specific locations. However, they can be expensive, time consuming and sometimes imprecise, making it difficult to continuously monitor soil properties over space and time. Applications worldwide have prompted the development of more time- and cost-efficient quantitative approaches to soil analysis that complement conventional laboratory techniques (Viscarra Rossel et al., 2010).

Diffuse Reflectance Infrared Fourier Transform (DRIFT) soil spectroscopy is a non-destructive, fast and inexpensive method that can predict soil properties and constituents by linking measured soil spectral patterns to reference values, which are usually attained via conventional laboratory methods (Stenberg et al., 2010). Accurate predictions can be made due to underlying relations between measured spectral patterns and absorbance features of soil characteristics such as color and both mineral and organic constituents (Nocita et al., 2015). Organic and inorganic carbon as well as soil texture are commonly accurately predicted using spectroscopic modelling approaches (Viscarra Rossel et al., 2006; Wijewardane et al., 2018; Clairotte et al., 2016; Dangal et al., 2019). Model predictions for cation exchange capacity (CEC), exchangable $Ca^{2+}$ and $Mg^{2+}$, soil pH, and several others have also shown promising results (Guillou et al., 2015; Reeves and Smith, 2009; Madari et al., 2006; Viscarra Rossel et al., 2008).

Soil spectroscopy can be performed using different wavelengths in the visible (Vis), near-infrared (NIR) or mid-infrared (mid-IR) portions of the electromagnetic spectrum. The main advantage of mid-IR spectroscopy (frequency of $4000\,cm^{-1}$ to $400\,cm^{-1}$ and wavelength of $2500\,nm$ to $25\,000\,nm$) under laboratory conditions is that the spectra hold more information on the mineral and organic composition of soil because the fundamental molecular vibrations occur mostly in this range (Janik et al., 1998; Reeves and Smith, 2009). More specifically, mid-IR range differentiates more between vibrations of molecular bonds, in contrast to the vis–NIR, where absorptions are broader and have more overlap. The more distinct absorption peaks and spectral features allow for a better separation of the soils' inorganic and organic components. Hence, potentially more precise and targeted spectral inference of a broad variety of mineral and organic soils can be made using mid-IR measurements.

The slight disadvantage is that often more sample preparation is needed compared to samples measured in the vis–NIR range. For spectroscopy in the mid-IR, unlike in the NIR range for example, the soil has to be finely ground in order to optimize the



signal to noise ratio (Guillou et al., 2015). This, however, makes it especially efficient to use prepared (legacy) soil datasets for mid-IR spectroscopy models.

In the soil spectroscopy modelling community, most current research efforts are focusing on minimizing the differences in performance between local (e.g. location or field specific) models versus large-scale (e.g. national, continental or global) models. On the one hand, this may be because the choice of the statistical model itself only results in slight performance

variability depending on the complexity of the soils. Traditional chemometric approaches (e.g. partial least squares regression (PLSR); e.g. Janik and Skjemstad, 1995), machine learning (e.g. regression tree methods; e.g. Clairotte et al., 2016; Dangal et al., 2019) and deep learning (e.g. convolutional neural networks; e.g. Padarian et al., 2019a, b) have all been used fairly successful. On the other hand, the premise of these studies may have been due to the fact that in the past, large-scale models still tended to perform less accurate than small-scale models (Guerrero et al., 2016; Stevens et al., 2013). Two reasons for lower

accuracy are a higher soil heterogeneity across larger spatial scales, which leads to a higher variability of spectral patterns and the limitations of statistical models to deal with such variability. Another reason is inharmonious sample preparation, measurement protocols and instruments (Nocita et al., 2015). Therefore, local models used to predict soil properties for a specific location or region were initially favored (Wetterlind and Stenberg, 2010; Stevens et al., 2013; Guerrero et al., 2016; Sila et al., 2016; Viscarra Rossel et al., 2016), but these had to be re-calibrated using new samples and laboratory analysis for

every new region.

More recently, however, methods were developed to use large soil spectral libraries (SSL) to predict soil properties locally at new locations, further minimizing the time and expenses required for sampling and laboratory work. Currently, several countries have established SSLs using archived, legacy and new soil data, such as The Czech Republic (Brodský et al., 2011), France (Gogé et al., 2012; Clairotte et al., 2016), Denmark (Knadel et al., 2012), China (Shi et al., 2014), the U.S. (Dangal et al.,

2019), Brazil (Demattê et al., 2019) and Switzerland (Baumann et al., submitted). Continental, e.g. Australia (Viscarra Rossel et al., 2008) or Europe (Stevens et al., 2013), as well as global SSLs have also been established (Viscarra Rossel et al., 2016; ICRAF, 2020). The operational value of SSLs lies in the ability to pull representative information (either the actual soil spectra or learnt model "rules") from them, requiring less new local samples and laboratory analysis. These methods can be summarized as augmenting SSLs with local soil samples, or spiking (Shepherd and Walsh, 2002; Brown, 2007; Wetterlind

and Stenberg, 2010; Seidel et al., 2019), memory- or instance-based learning (Ramirez-Lopez et al., 2013; Gholizadeh et al., 2016), subsetting (Araújo et al., 2014; Lobsey et al., 2017) or transfer learning (Padarian et al., 2019a), which is the process of sharing intra-domain information and rules learnt by general models to a local domain (Pan and Yang, 2010).

In this study, we used the RESAMPLING-LOCAL, or RS-LOCAL algorithm developed by Lobsey et al. (2017) because it combines several advantages of all four methods listed above. RS-LOCAL is a data-driven method to subset a SSL using spectra

from local samples. The subset includes these local, or spiked samples for calibration and may thus be summarized as instance-based transfer learning. In two case studies in Australia and New Zealand (Lobsey et al., 2017), the reduction of the SSL by means of local performance-based selection (RS-LOCAL) gave better results than constraining the SSL feature space by spectral similarity (memory-based learning).



We chose to specifically focus on soil carbon (SC) from peat soils in our mid-IR spectroscopic modelling approaches using different datasets for several reasons. Firstly, scientists agree that SC is an indispensable soil property for assessing agricultural lands (e.g. Noellemeyer and Six, 2015). Secondly, Cardelli et al. (2017) pointed out that spectroscopic modelling has almost only been used for mineral soils, stating the need for soil spectroscopy of more diverse datasets that include organic soils. Thirdly, we argue that currently, organic soil samples are underrepresented in SSLs and that this is a problem because the agricultural use of drained organic soils, or peatlands, is subject of immense debate in multiple sectors of societies. On the one hand, drained organic soils belong to the most fertile agricultural areas (Ferré et al., 2018), especially due to their high SOM content and the release of plant nutrients during mineralization. On the other hand, drained peatlands are a major source of greenhouse gas emissions (e.g. Parish et al., 2008; Joosten, 2010; Leifeld and Menichetti, 2018), susceptible to wind and water erosion (Zobeck et al., 2013), enhance subsidence of agricultural parcels due to compaction and rapid mineralization and are prone to flooding (Leifeld et al., 2011). As a result, often what remains is a substrate consisting of a thin organic horizon above a geologic and/or water-logging substrate. These factors have made crop production on such locations increasingly expensive; expenses may include drainage renovation or adding allochtone sand to the soil among other measures (Ferré et al., 2018).

Due to ongoing discussion of optimizing the land use of drained organic soils between stakeholders with agricultural, socio-economic and environmental interests, there is a need for using the advantages of mid-IR soil spectroscopic modelling to quantitatively characterize these soils. It is unknown whether current SSLs can ultimately be used to make location-specific land use decisions, particularly for small-scale heterogeneous regions made up of a variety of mineral and organic soils. In the current soil spectroscopy literature, there is to our knowledge no study about partitioning a SSL using RS-LOCAL with mid-IR spectroscopy, especially for a specialized organic soils dataset. Unlike Padarian et al. (2019a), who demonstrated the application of transfer learning at a continental scale, this study looks into the application of transfer models from a national to local scale, specifically for peat soils.

The aim of this study is to compare mid-IR spectroscopic modelling approaches of SC from peat soils using different datasets: 1) a local dataset specifically from drained peatlands; 2) the Swiss SSL spiked with local samples; and 3) RS-LOCAL subsets containing local and representative SSL samples. The goal is to develop both an accurate modelling strategy for predicting SC ranging between $1\%$ to $52\%$ and efficient calibration sampling scheme to minimize the number of new samples required.

## 2   Soil Data: The Swiss SSL and the HAFL Dataset

The current Swiss SSL in the mid-IR range consists of 3723 topsoil ($0\,\mathrm{cm}$ to $20\,\mathrm{cm}$) samples from 1094 locations from the Biodiversity Monitoring program (BDM; e.g. FOEN, 2018; Meuli et al., 2017) and 572 samples from 71 locations from the National Soil Monitoring Network (NABO; Table 1 and Figure 1; e.g. NABO, 2018; Gubler et al., 2015). The Swiss SSL is described in full detail in Baumann et al. (submitted).

Given the relatively low representation of organic soils and their organo-mineral diversity in the SSL, we introduce a dataset from the Bern University of Applied Sciences, School of Agricultural, Forest and Food Sciences (HAFL), which was set up





using a purposive sampling design to specifically study drained peatlands and organic soils. This local "HAFL" dataset ($n =$ 116) contains soil samples from between $0\,\mathrm{m}$ to $2\,\mathrm{m}$ depth from a range of natural as well as disturbed histosols from 26 different locations in the "Seeland" and "St. Galler Rheintal" regions of Switzerland (Figure 1; IUSS Working Group WRB, 2014). These samples originate from pedogenetic undisturbed and waterlogged organic horizons, or substratum, mineralized organic horizons under agricultural use, horizons with sandy or calcareous substrate material or horizons containing a mixture of these characteristics.

**Table 1.** Summary statistics of all samples in the HAFL dataset, in this study also referred to as the local dataset, and the BDM and NABO datasets. The latter two together constitute the current Swiss SSL, which contains carbon reference measurements from 4295 samples from 1150 different locations (Baumann et al., submitted).

| Dataset | Type | Locations | Samples (n) | Total carbon [%] | | | | | |
|---------|------|-----------|-------------|------|---------|--------|------|---------|------|
| | | | | Min. | 1st Qu. | Median | Mean | 3rd Qu. | Max. |
| HAFL | Local | 26 | 116 | 1.41 | 11.73 | 25.45 | 24.76 | 35.17 | 52.23 |
| BDM | Swiss SSL | 1079 | 3723 | 0.12 | 2.73 | 4.15 | 5.49 | 6.48 | 58.34 |
| NABO | Swiss SSL | 71 | 572 | 1.12 | 1.97 | 3.260 | 3.97 | 4.70 | 27.32 |

## 3 Methods

### 3.1 Mid-IR Soil Spectroscopy Measurements and Pre-Processing

All samples were dried, sieved ($< 2\,\mathrm{mm}$) and finely ground to optimize the signal to noise ratio (Guillou et al., 2015). The samples were measured with a VERTEX 70® FT-IR Spectrometer with a High Throughput Screening Extension (HTS-XT) from Bruker Optics (Massachusetts, USA). We used a spectral range of 7500–600 $\mathrm{cm}^{-1}$ and a spectral resolution of 2 $\mathrm{cm}^{-1}$ so that each spectrum comprised of reflectance values at 6901 wavelengths. On each 24-well plate, a fixed gold panel as the reflectance background, three NABO standards and two subsamples of 10 different samples were measured using the HTS-XT extension. This means that ten samples with two different measurements were analyzed per plate, improving the signal-to-noise ratio by averaging the two measurements. Reflectance spectra were transformed to apparent absorbance and recorded as such. OPUS® software was used for correcting atmospheric water and $CO_2$.

We tested several pre-processing steps in order to increase the information content for modelling and reduce collinearity between consecutive wavelengths. What follows are only the combination of parameters we ended up using. We used a Savitzky-Golay (SG) filter with a first derivative and second-order polynomial (Savitzky and Golay, 1964) in combination with a window size or resolution of 35 points, which is the equivalent of 70 $\mathrm{cm}^{-1}$. From these, we selected every $8^{th}$ variable to reduce collinearity and redundancy among predictors. This resulted in 209 variables between 634 $\mathrm{cm}^{-1}$ and 3962 $\mathrm{cm}^{-1}$, which formed the predictors for subsequent modeling. We used the "simplerspec" package for the R statistical language for reading spectra and metadata from Bruker OPUS® binary files into an R list, gathering spectra into a list column data structure,





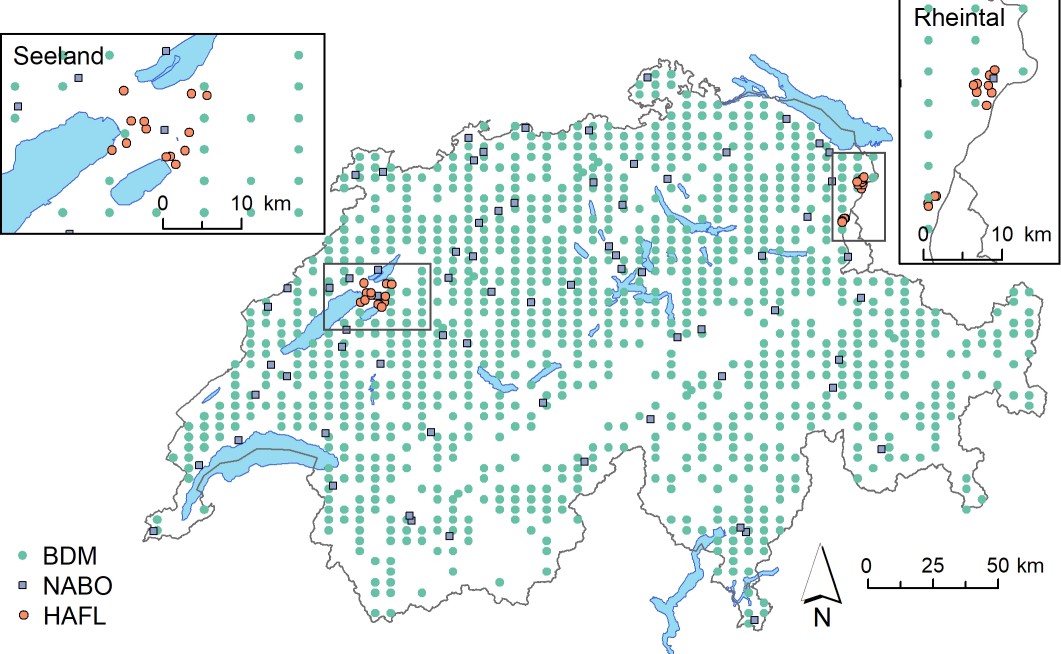

**Figure 1.** Spatial grid of BDM locations (green dots), NABO long-term monitoring locations (gray squares) and the purposive sampling design of peatlands in the "Seeland" and "St. Galler Rheintal" regions, which constitute the HAFL locations (orange dots) (NABO, 2018; Gubler et al., 2015; FOEN, 2018; Meuli et al., 2017; Baumann et al., submitted).

resampling spectra to new wavenumber intervals, averaging spectra of replicate scans and pre-processing the raw spectra with the parameters described above (Baumann, 2020; R Core Team, 2019).

### 3.2  Reference Chemical Analysis

In order to guarantee that reference data for all mid-IR spectroscopy models was measured using the same standard soil chemical analysis methods, we prepared and measured SC in the local HAFL dataset using the same procedures as for the
Swiss SSL (Baumann et al., submitted). Briefly, all the dried, sieved (< 2mm) and finely ground samples were measured for total carbon content by dry combustion using the CHN628 Series Elemental Determinator® from the Laboratory Equipment Corporation (LECO Corp., St, Joseph, MI, USA). We used a soil standard sample with a mean total carbon content of 2.372 %. In order to compare measurement accuracy and accordance between the two different CHN628 Series Elemental Determinators used, representative samples were selected using a two-step process. The data was separated into two clusters using the K-
Means clustering algorithm (Hartigan and Wong, 1979), followed by using the Kennard-Stone (KS) algorithm for each cluster separately (Kennard and Stone, 1969). The KS algorithm is a deterministic approach that uses Euclidean or Mahalanobis distance to select a set of samples uniformly distributed in principal component (PC) space (Kennard and Stone, 1969).



### 3.3 Spectroscopic Modelling

We resampled the data using a 5 times repeated, grouped by location, 10-fold cross-validation (CV) for all of our models to
determine the optimal number of components in model tuning as well as evaluating the model performance using the hold-out
samples. The predicted carbon content was calculated in each model using the hold-out values of the measured, pre-processed
and averaged mid-IR spectra. For the final models, the calculated average (mean) predictions over these five repeats with the
chosen number of components are shown. To avoid overfitting, we used the "one standard error" rule: instead of choosing the
tuning parameter associated with the best performance, we chose the simplest model within one standard error (SE) of the
empirically optimal model (e.g. Hastie et al., 2009).

We used partial least squares regressions (PLSR; e.g. Wold, 1975; Wold et al., 1983, 1984, 2001) to predict SC. We tuned
the PLSR using 1 to 10 components and the final model for number of components was chosen according to the "one SE" rule.
We also fit Cubist models tuned with different number of committees (5, 10 and 20) and neighbors (2, 5, 7, 9), but these only
slightly changed model predictions and for simplicity are not included in the results presented here.

All spectroscopy models were evaluated for their performance using the root mean squared error (RMSE) to assess accuracy,
the bias to assess the mean error and the mean square error skill score ($SS_{mse}$), which is commonly interpreted as $R^2$, to assess
the model fit (Nussbaum et al., 2018; Wilks, 2011). One advantage of repeating CV is that model imprecision can easily be
assessed for each prediction $\hat{Y}$ using the standard deviation (SD) and mean $\bar{Y}$ of the predictions across 5 repeats, respectively.
In this study, the SD is shown as error bars for each prediction $\hat{Y}$ and were also calculated for each overall summary statistic
while assessing model performance.

### 3.4 Calibration Schemes

#### 3.4.1 Local Models

Spectroscopy becomes time and cost efficient when you minimize the amount of laborious chemical reference analysis. There-
fore, it makes sense to split the local HAFL data into a calibration and validation subset (Figure 2). In this manner, the validation
can be used to determine how many samples are needed to accurately and precisely calibrate a model. We selected $n = 15$,
20, 25, 30, 40, 50 and 58 representative local HAFL calibration samples by using the KS algorithm to the first 5 PCs. Models
cannot be calibrated using less than 15 soil samples in a 5 times repeated, grouped by location, 10-fold cross validation and
so results were not comparable and hence not shown. Calibrating a model with so little data is also highly questionable. These
selected samples were then used to calibrate iterations of PLSR models (Figure 2). In an application of the method described,
reference data would only have to be measured for the selected samples used for calibration. Each calibrated model iteration
was validated using the remaining local samples of each iteration.





### 3.4.2 SSL Spiked Models

In the next step, we utilized the Swiss SSL in iterations of model calibrations to see if predictive performance can be improved while further reducing the number of new local samples needed for reference analysis (Figure 2). Also, we expected a large
amount of additional data from the SSL to improve model robustness and reliability (Lobsey et al., 2017). With the help of all SSL samples containing carbon reference data ($n = 4295$), we were able to include iterations of PLSR calibrations spiked with as few as $n = 3, 5, 7$ and $10$ local HAFL samples. Further iterations with the same $n = 15, 20, 25, 30, 40, 50$ and $58$ local HAFL samples as for the local models were also calibrated. Just as with the local models, each iteration was validated with the remaining local HAFL samples not used in model calibration ($116 - n$).

### 3.4.3 Models using RS-LOCAL Subsets

In order to further increase the efficiency of soil spectroscopy, we tested whether representative subsets of the SSL using the RS-LOCAL algorithm improved the accuracy of predicting SC of the local HAFL samples (Figure 2). The RS-LOCAL algorithm was used to data-mine the SSL for samples suitable for local or location-specific calibrations according to similarities of spectral signatures between the local HAFL and SSL soil samples (Lobsey et al., 2017). Local HAFL samples were selected in the same
manner as in local and SSL spiked models, resulting in iterations of the same samples as before (Figure 2). This variable was defined in the RS-LOCAL algorithm as $m$ (Lobsey et al., 2017); so in our case, $m = 3, 5, 7, 10, 15, 20, 25, 30, 40, 50$ and $58$ for each respective iteration. RS-LOCAL used $m$ data to resample, evaluate and then remove irrelevant data from the SSL so that only the most appropriate data for deriving a local calibration remain in a new SSL subset $K$. $K$ and $m$ together formed an RS-LOCAL dataset, which was used for a calibration. In addition to the SSL and $m$ data, three parameters were needed for
RS-LOCAL (Lobsey et al., 2017):

- $k$: the number of SSL samples randomly selected in the resampling step, and also the target number of SSL samples returned by the algorithm

- $b$: the number of times each sample in the SSL was tested, on average, in each iteration of the algorithm

- $r$: the proportion of SSL samples removed in each iteration of the algorithm

In order to optimize the tuning parameters, we performed a full factorial combination of $k$ (50, 100, 150 and 300), $b$ (40, 50 and 60) and $r$ (0.05 and 0.01) based on recommendations of the developers (Lobsey et al., 2017). As in the other scenarios, the remaining local HAFL samples not in $m$ were used for model validation for each respective iteration ($116 - m$).



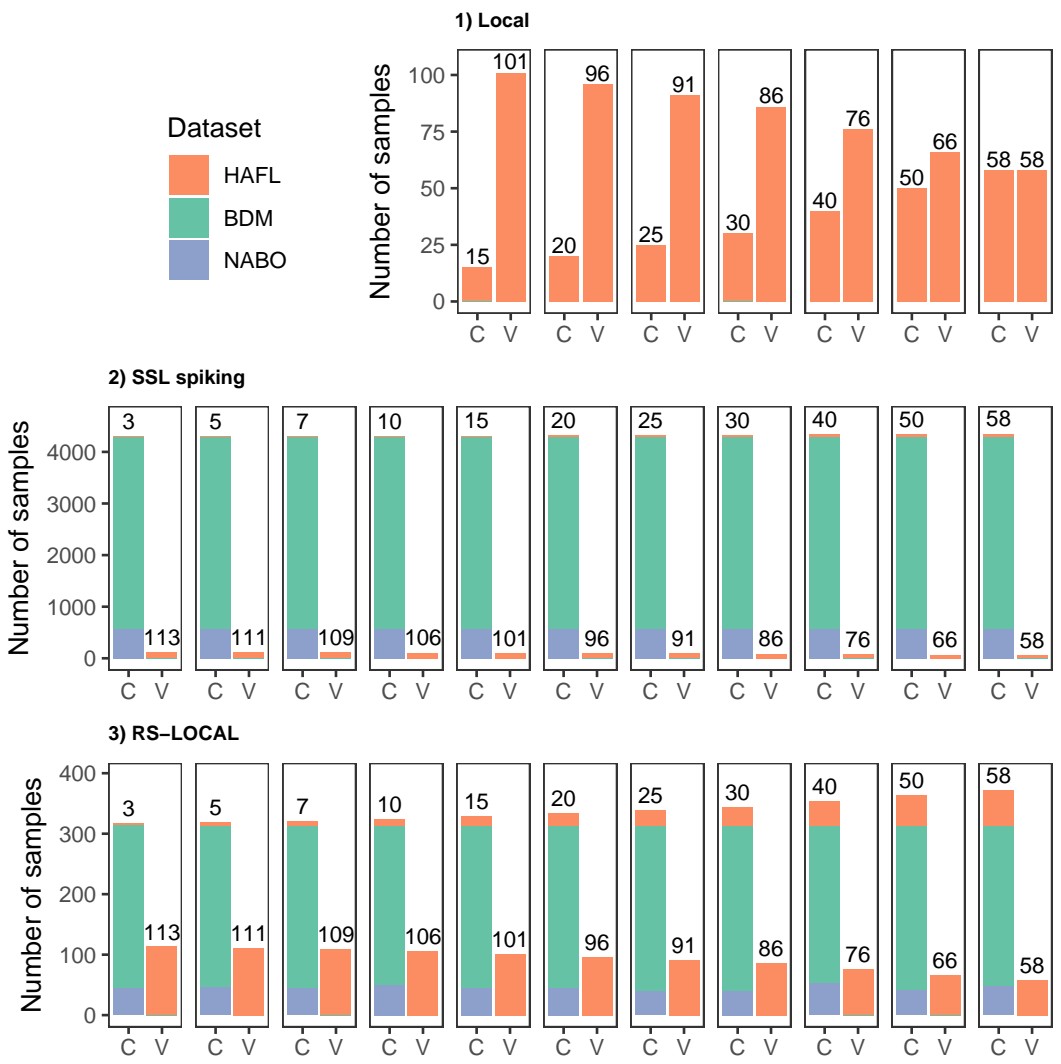

**Figure 2.** The number of samples used in each calibration (C) and validation (V) scheme using HAFL, BDM and NABO datasets for 1) local, 2) SSL spiking and 3) RS-LOCAL approaches. The same local HAFL samples not used in calibration were always used for validating the modelling approaches. Note different scales of the y-axis.





# 4   Results

## 4.1   Analysis of Local HAFL Spectra

The raw and pre-processed measured mid-IR spectra of the local HAFL soil samples ($n = 116$) were clearly distinct in relation to the SC content, ranging from $1\,\%$ to $52\,\%$ (Figures 3a & b). Mineral and organic soil samples showed different absorbance patterns in both the raw and pre-processed mid-IR spectra. Pre-processed absorbance values showed a clear pattern at specific wavenumbers according the the SC content, such as around $2000\,\mathrm{cm}^{-1}$, where soils with a low carbon content had a higher absorbance than soils with a high carbon content.



**Figure 3.** The raw (a) and pre-processed (b) mid-IR spectra of all the local HAFL soil samples ($n = 116$) colored by total carbon content [%].





A PCA of the pre-processed spectra of the local HAFL samples clearly revealed a variance in distribution of the soil samples related to the SC content (Figure 4). The first two PCs together explained $53.5\%$ of the total variance in the pre-processed spectra. In the example shown, 20 representative samples were selected for calibrating a local model using the KS algorithm and the rest of the data were used for validation ($n = 96$). We also compared the similarity of soil samples from different depths at the same location location by coloring the first two PCs by location (Appendix A). However, the pre-processed spectra from

the same locations generally showed little similarity; there was no distinct pattern as there was for the SC content.

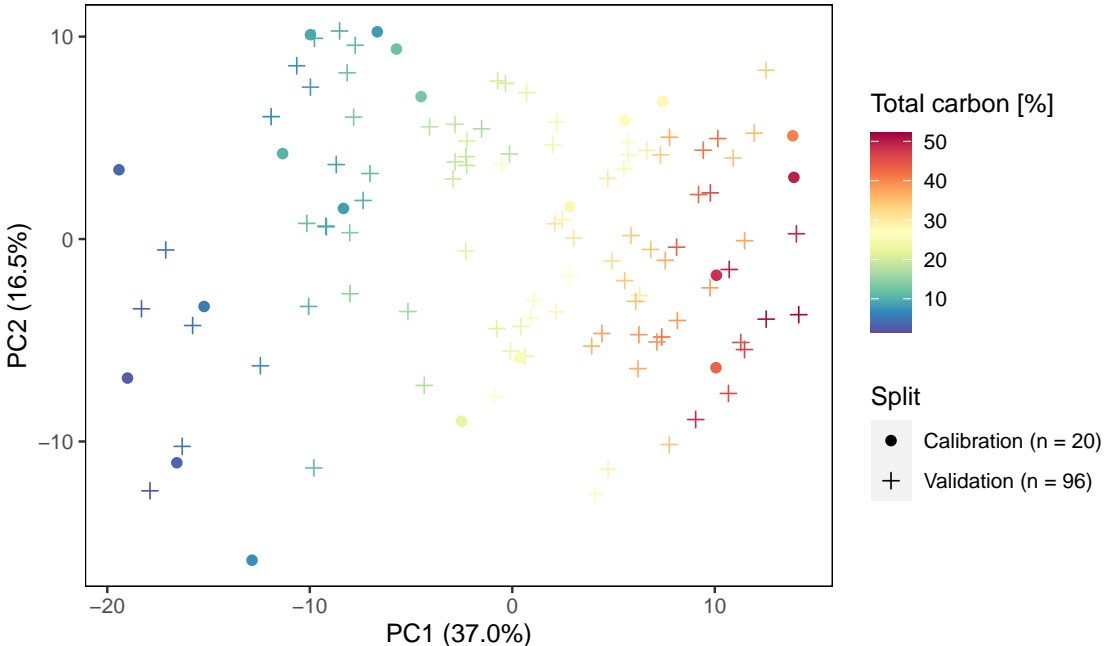

**Figure 4.** PC1 vs. PC2, computed via scaled and centered PCA, of the pre-processed local HAFL soil spectra ($n = 116$) colored by total carbon content [%]. The axis labels show the relative amount of variance explained [%] in parentheses. In this example, $n = 20$ representative samples were selected for calibrating a local model using the KS algorithm and the rest of the data were used for validation ($n = 96$).

## 4.2  Comparing Local HAFL to SSL data

When comparing the three datasets, we found that the local HAFL dataset showed a different SC distribution and covered a different range of soil variability than the Swiss SSL (Table 1 and Figure 5). The relatively small HAFL dataset originating from peaty soils had a uniform continuous distribution whereas the BDM and NABO data had a positively skewed distribution with

regard to SC (Table 1). Although the BDM dataset contained the highest value of SC, the overwhelming majority of soil samples in the Swiss SSL originated from mineral soils. In contrast, more than half of the HAFL samples ranging from $1\%$ to $52\%$ were classified as organic soils. The first and second PCs – which together covered $40.3\%$ of the total variance – also revealed a clear overlap of pre-processed mid-IR absorbance variance for the BDM and NABO datasets (Figure 5). There was less overlap



in the variance of the pre-processed mid-IR absorbance values of the Swiss SSL and the HAFL dataset. In the example shown,
122 samples, of which 20 are the local HAFL samples, were used as a RS-LOCAL subset to calibrate a model. The remaining
BDM and NABO samples in the SSL were not used in this example ($n = 4193$). The remaining local HAFL samples were used
to validate this model ($n = 96$).

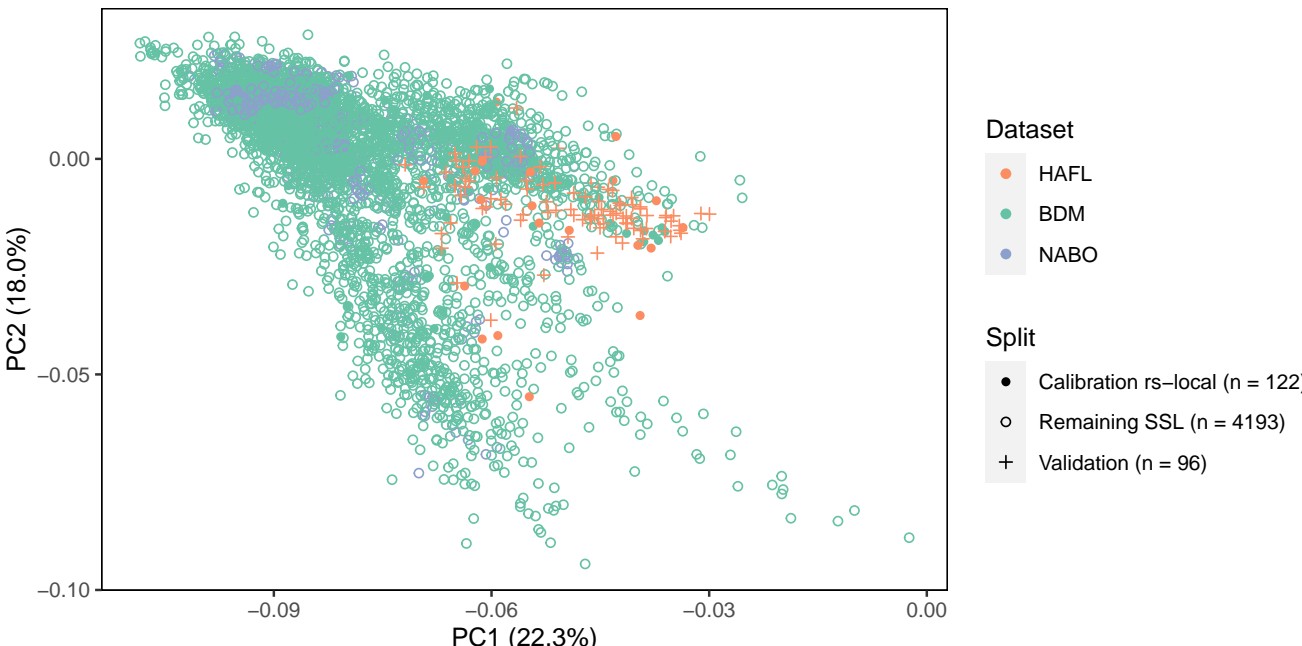

**Figure 5.** PC1 vs. PC2 of the pre-processed soil spectra of the local HAFL dataset ($n = 116$) and all of the BDM and NABO datasets for
which reference carbon measurements were available ($n = 4295$). Here, the PCs were computed via unscaled and uncentered PCA since
it better showed the data distribution than a scaled and centered PCA. The axis labels also show the relative amount of variance explained
[%] in parentheses. In this example, the same representative local HAFL samples ($n = 20$) selected in Figure 4 were used by RS-LOCAL to
subset representative samples from the SSL ($n = 102$), which were used together for model calibration ($n = 122$). The remaining local HAFL
samples ($n = 96$) were used for validation.

### 4.3  Predicted SC Using 1) Local, 2) SSL Spiking and 3) RS-LOCAL Subsets

We predicted SC content ($\hat{Y}$) of the HAFL, BDM and NABO datasets using mid-IR soil spectroscopy and PLSR models and
compared them to the reference chemical measurements ($Y$) (Figures 6 & 7). For models using an RS-LOCAL subset, we found
best overall validation results using $k = 100$, $b = 50$ and $r = 0.05$ and therefore chose these parameter values for all results
shown here. Decreasing $r$ to 0.01 had little influence on model performance and greatly increased computation duration. All
models were then validated using the remaining local HAFL samples ($116 - n$) (Figures 6b & 7b).





All models used for calibration showed a high fit ($R^2 > 0.9$) and low overall bias ($\approx 0$) (Figure 6a). However, the accuracy
of the spiked SSL calibration decreased and bias increased in the upper range of SC. This effect was dampened in the overall
summary statistics due to the overwhelming majority of samples in the lower SC range. The local PLSR using only 20 samples
showed the lowest overall calibration accuracy (RMSE = 3.72 % total carbon).

Although all model validations showed a similar fit ($R^2 = 0.94 - 0.96$), bias and RMSE values differed among the three
scenarios (Figure 6b). The validation of the local PLSR had the lowest bias (0.5), but the validation of the RS-LOCAL subset
had the highest overall accuracy (RMSE = 2.78 % total carbon). The validation of the spiked SSL scenario had the highest bias
and lowest accuracy. The error bars representing the standard deviation of predictions across five resampled repeats showed
that there was highest model uncertainty in predictions across repeats in the local calibration and validation.

As with 20 local samples, validation statistics also show good performance of PLSR models using RS-LOCAL subsets
that contained as few as 5, 7 and 10 local samples ($R^2 = 0.94 - 0.96$, bias $= -0.12 - 1.51$, RMSE = 2.78 % to 3.5 % total
carbon; Figure 7). Validations with 5 and 10 local samples are very similar, while validation of the model with 7 local samples
performed the worst among the three. Model performance of the RS-LOCAL calibration iterations themselves, with 5, 7 and 10
local samples, did not vary considerably.

We compared overall goodness of fit using $R^2$, bias and accuracy using RMSE for different scenarios of 1) local, 2) SSL
spiking and 3) RS-LOCAL subsets depending on the number of local HAFL samples included in each PLSR (Figure 8). Each
calibrated model was again validated with all remaining HAFL samples not included during calibration. All model validations
showed a high goodness of fit ($R^2 \geq 0.94$), except when the calibration only included 3 local samples in a RS-LOCAL subset.
Purely local models showed the lowest bias, regardless of the number of samples used during calibration. The bias of validations
of spiked SSL models only lowered slightly as the number of local samples was increased and remained large overall ($> 3$).
Bias was reduced significantly in validated models of RS-LOCAL subsets when at least 5 local samples were included and
continued to decrease with increasing number of local samples, although models with 5 and 10 samples stand out with very little
bias. Model accuracy (RMSE) varied considerably between calibrations and validations when SSL samples were used during
calibration. There was less of a difference between calibration and validation in local models, where the RMSE decreased with
increasing number of local samples in model calibration. As with model bias, the RS-LOCAL subsets again showed a threshold
or minimum of 5 samples in order for the RMSE of model validations to lower significantly. Model validations of local and
RS-LOCAL subsets showed very similar accuracy overall (RMSE $\approx 3$ % total carbon).





**Figure 6.** Predicted ($\hat{Y}$) vs. observed ($Y$) total carbon content [%] by calibrating a PLSR with 20 local HAFL samples (a) and validating it with the remaining local HAFL samples ($n = 96$) (b). A PLSR was calibrated using only 20 local HAFL samples (1), adding the entire SSL to the 20 local HAFL samples (2) and using a RS-LOCAL subset of the SSL also containing the 20 local samples. The error bars signify the SD and where none are present, the components remained identical across 5 repeats and thus SD = 0.





**Figure 7.** Predicted ($\hat{Y}$) vs. observed ($Y$) total carbon content [%] by calibrating a PLSR of a RS-LOCAL subset with 5, 7 and 10 local HAFL samples (a) and validating it with the remaining local HAFL samples ($n - 116$) (b). The error bars signify the SD and where none are present, the components remained identical across 5 repeats and thus SD $= 0$.




**Figure 8.** $R^2$, bias and RMSE values in % total carbon (RMSE in logarithmic scale) vs. the number of local HAFL samples used in local HAFL (1), local combined with SSL (2) and RS-LOCAL subsets of the SSL (3). All models were fit using PLSR, adjusting the number of local HAFL during calibration of each scenario and validated with the remaining local HAFL samples, respectively. Error bars represent the SD between resampled repeats and where not present, there was no variation in the respective summary statistic across five repeats. Dashed lines between data points do not represent real measurements and are only for visual guidance.





## 5 Discussion

We found that firstly, mid-IR spectra can be used to predict SC up to $52\,\%$ with $R^2 \geq 0.94$, negligible bias and RMSE $= 2.8\,\%$ to $3.4\,\%$ total carbon using validated local PLSR models. Secondly and most importantly, time-consuming and expensive field and laboratory measurements can be reduced for new locations when using a SSL and RS-LOCAL. In our study, only 5 local

HAFL samples were required in a RS-LOCAL subset to achieve similar validation performance as with at least 50 local samples in a local model ($R^2 = 0.95$, bias $\approx 0.5$ and RMSE $\approx 2.9\,\%$ total carbon; Figure 8). This is a major improvement to local models without a SSL because it not only reduces field and laboratory expenses, but also because no reliable model can be calibrated using so little data. Furthermore, a SSL subsetting method such as RS-LOCAL combined with a simple model such as PLSR are easy to understand and require little computational power compared to alternative machine- or deep learning

approaches (e.g. Padarian et al., 2019a, b).

### 5.1 Spectral Patterns Reflect Mineral and Peat Composition

The measured and pre-processed mid-IR soil spectra and PCA results of all local HAFL samples revealed high correlation between the spectral absorbance values and a broad range of SC content (Figures 3 & 4). According to past studies that assessed variable importance, we assumed that soil color, texture and mineralogical and organic composition influence mid-

IR spectral absorbance the most (Madari et al., 2006; Bornemann et al., 2010; Calderón et al., 2011). SOC, for example, is known to be related to a variety of bands that represent absorptions due to organic molecules such as proteins with C——O, C══O and N——H bonds (Viscarra Rossel and Behrens, 2010). Local HAFL samples containing both a high amount of organic compounds as well as carbonates created distinct absorbance bands around $1450\,\mathrm{cm}^{-1}$, $1460\,\mathrm{cm}^{-1}$, $2855\,\mathrm{cm}^{-1}$ and $2930\,\mathrm{cm}^{-1}$ for aliphatic C-H constituents (Madari et al., 2006) or around $1320\,\mathrm{cm}^{-1}$ for hydroxyl groups bonded to carbon (C-

O-H) (Bornemann et al., 2010) and around $2500\,\mathrm{cm}^{-1}$ for carbonates (Calderón et al., 2011). Future studies should investigate whether overlapping of spectral signals for organic and mineral components increases at high SC concentrations, implying that spectral absorbance patterns above a certain threshold of SC no longer differentiate substantially.

The correlations between the mid-IR soil spectra and SC of a range of peat samples present the opportunity for future studies to analyze and possibly monitor peat composition. For example, the authors of Douglas et al. (2019) used vis-NIR spectroscopy

to rapidly detect alkanes and polycyclic aromatic hydrocarbons, which are also relevant constituents in peat. In addition, since N content can also be predicted using mid-IR spectroscopy (Viscarra Rossel et al., 2008; Ma et al., 2019), this technique should also be able to monitor changes in peat according to the C/N ratio. With an increasing stage of peat mineralization, the C/N ratio becomes smaller because carbon escapes as $CO_2$ whereas N is initially mostly stored in the microbial biomass and ultimately over $95\,\%$ is stabilized in organic bonds (Blume et al., 2010; Krüger et al., 2015). Finally, note that a broad

range of recent studies have already analysed peat composition using vis-NIR and mid-IR FTIR spectroscopy, another type of vibrational spectroscopy where light passes through the sample instead of reflecting as in DRIFT spectroscopy. For example, it has been used to predict the effects of peatland type and drainage intensity on SOM and peat decomposition (Biester et al., 2014; Heller et al., 2015), the OM quality in regenerating peatlands (Artz et al., 2008), the humification degree of peat soil





(Sim et al., 2017) and the chemical composition and decomposability of arctic tundra SOM (Normand et al., 2017; Matamala
et al., 2019).

## 5.2 RS-LOCAL Improves Model Performance and Increases Efficiency

The RS-LOCAL approach using representative local and SSL samples was found to be the best of the three compared ap-
proaches. It significantly reduces the amount of reference measurements that need to be made at new locations to 5 samples.
In addition, the RS-LOCAL approach helps remove the strong bias of spiked SSL calibrations (Figures 6 & 8), increasing the
under-estimated predictions of SC in the upper range. Finally, the additional samples provided from the SSL also reduce uncer-
tainty of SC predictions of resampled repeats in the PLSR models, as can be seen from the smaller error bars of the residuals.
It can be expected that especially predictions of SC in the lower range became more accurate and precise considering the
distribution of SC in the SSL (Table 1 & Figure 5), although this cannot be directly inferred from our results.

We postulate that SC prediction accuracy of organic soil samples using SSL-derived models may be improved in future
studies by adding more peat soil data to the Swiss SSL (Baumann et al., submitted), specifically samples at different decompo-
sition and mineralization stages. One of the most important characteristics of a high-quality SSL is that it contains the highest
possible variation of soil characteristics within its designated area (Viscarra Rossel et al., 2016). The SC distribution and the
spectral principal component space of the HAFL compared to the BDM and NABO samples showed that the soil variability of
the HAFL samples is only marginally covered by the current SSL (Table 1 & Figure 5).
The use of our modelling approach, PLSR of RS-LOCAL subsets, to predict soil properties at new locations for future studies
and applications depend on the level of accuracy and precision needed. For organic soils on a farm or landscape level, an
accuracy of approximately 2 % to 3 % total carbon is suitable for quantifying SC. However, this range of accuracy is not
useful for mineral soils, which, in Switzerland for example, contain average (mean) topsoil SOC concentrations of 2 % on
arable locations and 2.5 % on temporary grassland locations (Leifeld et al., 2005). Nonetheless, calibrating a model with a
RS-LOCAL subset containing only 3 local HAFL samples resulted in samples below 20 % SC and respective RMSE = 0.6 %.
Hence, for samples with a lower SC range, accuracy increases. For future studies using spectral predictions over such a high
range of values, it may be useful to calculate the error for specific increments along the entire range of values, for example by
increments of 10 % SC. Furthermore, calculating weighted-RMSE values allows an error analysis directly dependant on SC
content. Such weighted-averaging approaches have often been used in global spatial models (Wieder et al., 2014).
To our knowledge, very few studies have predicted organic soils up to 52 % total carbon using soil spectroscopy, and none
have done so with mid-IR (as opposed to Vis-NIR) or without splitting model calibration for mineral and organic soils. Nocita
et al. (2014) predicted SOC for croplands, grasslands, woodlands and organic soils separately of about 20'000 samples from
the Land Use/Cover Area frame Statistical Survey (LUCAS) across Europe using Vis-NIR spectroscopy. For the model using
only organic soil data with a range of 12.0 % to 58.68 % SOC, predictions were less accurate (RMSE = 5.114 % SOC) than in
this study (Nocita et al., 2014).

One advantage of the RS-LOCAL-PLSR approach used here is that the statistical modelling is simple and produces easy
to understand models compared to other transfer and deep learning approaches (Padarian et al., 2019a, b). Using spectral- or



model-based information from SSLs by spiking, subsetting and memory, instance and transfer learning may even be beneficial for parts of the world lacking legacy soil data or funding to establish their own SSL. In other words, a SSL of one region may
be used to predict locally for another region that does not have a SSL.

However, there are still some drawbacks. As mentioned by Padarian et al. (2019a), spiking and subsetting are dependent on the size of local and global datasets and may still bias the predictions towards the local dataset rather than fully using valuable global information, which generates less robust models. Although transfer learning of model "rules", or network weights, showed promising results on a continental scale (Padarian et al., 2019a), it has not been tested when transferring
national spectral knowledge to a field scale. Furthermore, although spectroscopy predictions were used as input point data in the Australian soil information system (Viscarra Rossel et al., 2015), it is still unknown how these predictions influence soil maps and associated uncertainties.

## 6 Conclusion

This study reveals that, if adequately mined, the information in a SSL is sufficient to predict soil carbon of a new study
region with very different soil characteristics. Whereas past spectroscopy studies mostly focused on mineral soils, these model validations of SC ranging between $1\%$ to $52\%$ show that using as few as 5 new samples in combination with RS-LOCAL and a SSL yields promising results. This approach decreases the time and cost of field and laboratory soil analysis, reduces the bias of large-scale spectroscopy or SSL spiked models and the uncertainty of small-scale local models. Including more organic soil samples in the Swiss SSL will make it more robust for future modelling applications (Baumann et al., submitted). This case
study for assessing SC in drained peatlands under agricultural management shows that an operative SSL is useful for scaling up quantitative soil information over space and time.



*Code and data availability.* The datasets and code to reproduce the results of this manuscript are available upon request.

*Author contributions.* All authors contributed to the scientific research in this project and read through, edited and provided advice on improving the manuscript. Anatol Helfenstein measured all SSL and HAFL samples using mid-IR spectroscopy, performed the data preparation
and analysis, wrote most of the R code and wrote the manuscript. Some of the R code was written by Philipp Baumann and RS-LOCAL was implemented in R by Raphael Viscarra Rossel. Andreas Gubler provided the maps of sampling locations. Stefan Oechslin collected and prepared the HAFL soil samples. Johan Six helped with the interpretation of the results and improving the writing.

*Competing interests.* The authors declare no competing interests.

*Acknowledgements.* We would like to thank the authors in Baumann et al. (submitted) for the establishment of the Swiss SSL. We are grateful
to Daniel Wächter and Reto Meuli from the Swiss national soil monitoring group at Agroscope for the consent to access, quality control and data management of both milled soil samples and the database with reference measurements of the National Soil Monitoring Programme (NABO) and Swiss Biodiversity Monitoring programme (BDM) datasets. Many thanks to Stéphane Burgos and Madlene Nussbaum from the Bern University of Applied Sciences (BFH), who gave advice on the spectroscopic models and shared their knowledge of (drained) peatland soils. We would also like to thank Raphael Viscarra Rossel and Juhwan Lee for the valuable input and time spent at the Commonwealth
Scientific and Industrial Research Organisation (CSIRO) in Canberra, Australia. Finally, we express our gratitude to the Swiss Federal Office of the Environment for commissioning and funding the soil analyses conducted within the BDM.




## Appendix A: Variance of Local HAFL Samples Over Depth

Local HAFL soil samples from different depths of the same location were diverse. This may be due to pedogenetic formation conditions unique to peatlands near bodies of water as well as anthropogenic influence. The "Seeland" and "St. Galler Rhein-

tal" regions are characterised by an extreme diversity of intact peat, decomposed and mineralised peat, calcareous lacustrine sediments and fluvial sand, silt and clay deposits depending on past river flow conditions (Bader et al., 2018; Burgos et al., 2018). Anthropogenic influence such as changing the course of and channeling rivers, lowering lake and groundwater tables and draining peatlands further complicate soil characterisation. These conditions create a mosaic of extremely heterogeneous soil characteristics that vary vertically depending on soil depth as strongly as they vary horizontally across the entire three

study areas that are part of the HAFL dataset (Figure A1).

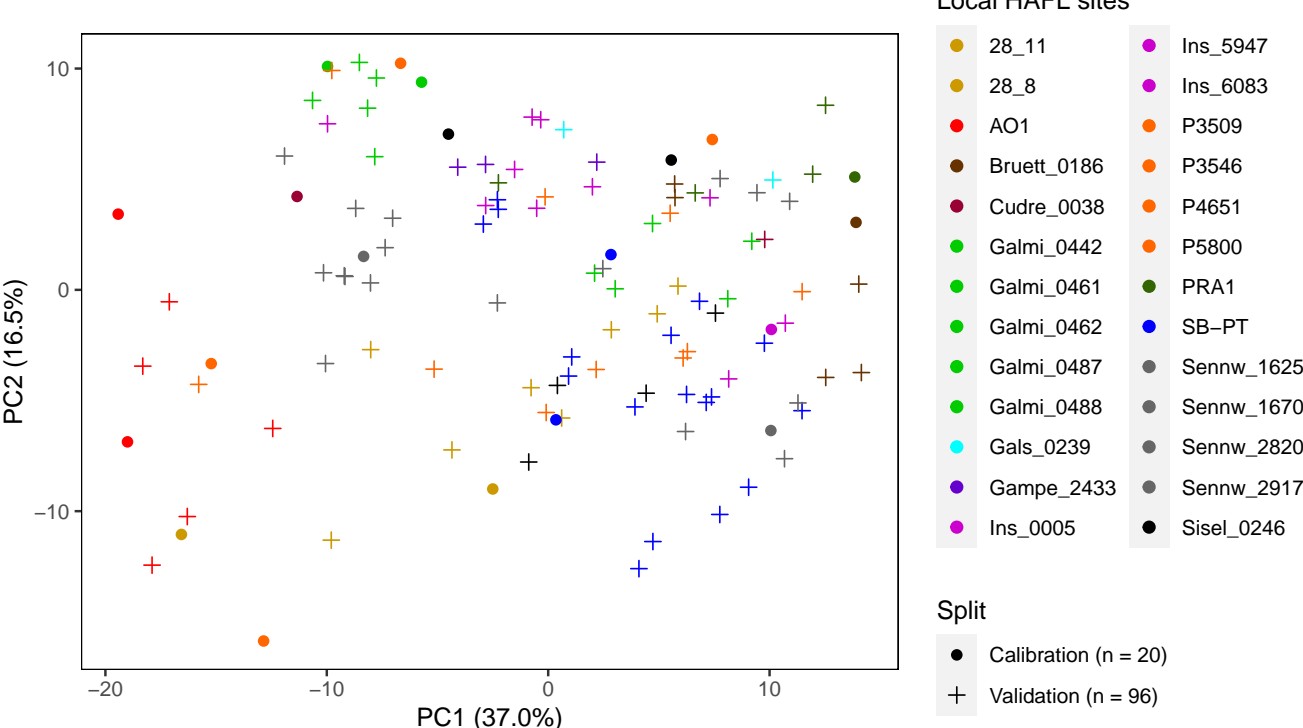

**Figure A1.** PC1 vs. PC2, computed via scaled and centered PCA, of the pre-processed local HAFL soil spectra ($n = 116$) colored by location, in contrast to Figure 4. The axis labels show the relative amount of variance explained [%] in parentheses. In this example, $n = 20$ representative samples were selected for calibrating a local model using the KS algorithm and the rest of the data were used for validation ($n = 96$).





## Appendix B: Number of Chosen Components

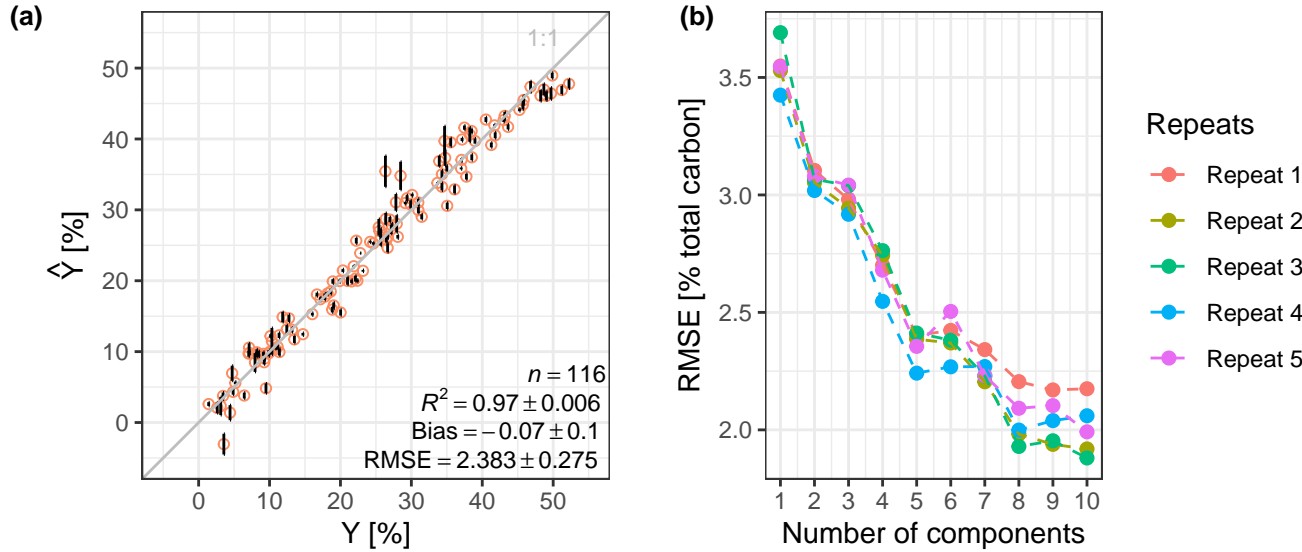

**Figure B1.** a) Predicted $\hat{Y}$ vs. observed $Y$ total carbon [%] of all local HAFL samples using a PLSR. b) Illustrative example of PLSR tuning, showing the calculated RMSE [% total carbon] vs. the number of components used in each repeat. This model used 7 components in the first and fifth repeat and 8 components in the second, third and fourth repeat according to the "one SE" rule.




## Appendix C: Geographical Position of Chosen Samples by RS-LOCAL

We mapped the locations from which RS-LOCAL selected samples from the SSL for calibration together with 20 local HAFL samples ($n = 334$; Figure C1). There appeared to be no spatial correlation between chosen RS-LOCAL locations and geograph-
ical distance from the local HAFL locations. In other words, locations chosen by RS-LOCAL suggest that spectrally relevant soil samples from the SSL to predict SC in local HAFL samples are not confined to nearby areas in terms of geographical distance. This may be linked to the heterogeneity of soils found in these drained peatlands: in between layers of organic soils, sampled soil layers also contained geologic substrate material from lacustrine carbonates, dense clay or fluvial sand depositions. We speculate that this soil and spectral diversity at local HAFL sampling locations may explain why RS-LOCAL even selected rele-
vant SSL samples originating from the Alps mountain range. This ultimately suggests that RS-LOCAL is able to use segments of soil spectra from a variety of similar but also dissimilar locations for prediction of new local soil samples.

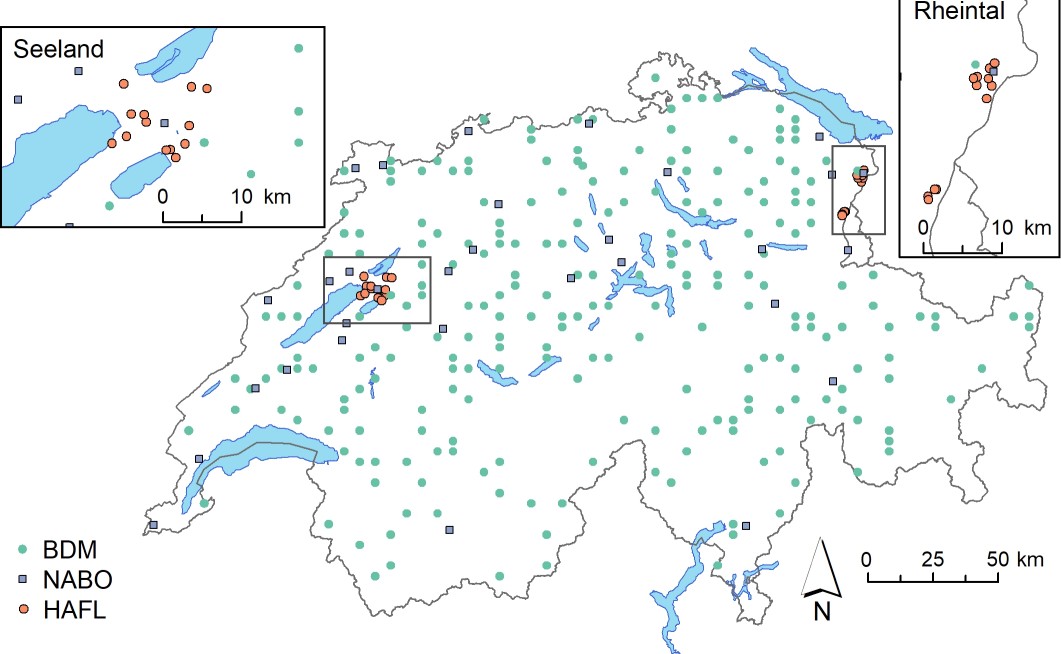

**Figure C1.** A map showing BDM (green diamonds), NABO (gray squares) locations from which RS-LOCAL subset samples ($n = 314$) for a RS-LOCAL calibration ($n = 334$). All HAFL locations (orange diamonds) are shown underneath BDM and NABO locations to allow a better focus on locations chosen by RS-LOCAL.



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
