# Peer review of "Quantifying soil carbon in temperate peatlands using a mid-IR soil spectral library"

_SOIL, 2020_

## Referee Comment (RC2)

**Referee's comments on soil-2020-93**

Title: Predicting soil carbon by efficiently using variation in a mid-IR soil spectral library

https://doi.org/10.5194/soil-2020-93

The topic addressed by the paper is interesting and definitely in the scope of the journal. The article deals with the use of mid-IR spectroscopy in conjunction with chemometrics by means of PLS regression to predict soil carbon content of organic soils collected from two peatland regions of Switzerland. More specifically, the authors propose a statistical and rational approach based on three datasets: i) local dataset from the two peatland regions, ii) national soil spectral library (SSL) spiked with local samples, and iii) subsets containing local and representative SSL samples.

The results of this research work are of interest for the scientific community, particularly those involved in the use of IR spectroscopy for the study and characterization of soils. The proposed approach allows promoting the use of SSL (national or regional) for local predictions while minimizing the number of local reference analyses required. Having said that there are several points that need to be fixed by the authors.

**General comments**

The authors do not satisfactorily explain the relevance of their choice to work on the calibration of total carbon prediction models rather than organic and/or inorganic carbon separately. In most research work, the organic form is given much more attention because of its central role in the functioning of soils.

The approach for the calibration of local models, by considering calibration set with data (15, 20, 25, 30 …) less or equal to that of the validation set, is statistically questionable. Moreover, how robust are these calibration models? What would happen if we made 5 random draws instead of the KS algorithm? The classical approach consists in in splitting a data set, taking 2/3 for calibration and 1/3 for validation. So why the authors did not go so far as to test this classical subdivision?

The authors used the following statistical criteria $R^2$, bias and RMSE to evaluate the prediction performance of spectral models in both calibration and validation. Most of the scientist involved in the field of soil spectroscopy also use RPD or even RPIQ (recommended when the distribution is not normal) criterion. I think the authors should provide them so that they are available to readers who would like to compare their results.

**Specific comments**

Title: Maybe it should refer to peatlands to make it a bit more explicit.

L. 116-119: The BDM database is made up of soil samples collected from topsoil (0-20 cm), is it the same for NABO samples?

L. 120: The organic soils are not sufficiently represented in the SSL. It would nevertheless be interesting to know their amount (or fraction) and their range of variation in SC content.

L. 122-123: the local HAFL consists of samples collected from 0 up to 2 m depth. I assume these are soil cores that were collected. In that case, what was the thickness of the soils taken for the reference analyses and what were the depths considered?

L. 168-169: No need to talk about Cubist models since the results are not included in this article.

L. 218-219: There is an ambiguity in this sentence:" where soils with a low carbon content had higher absorbance than soils with a high carbon content". According to the pre-processing carried out by the authors, Figure 3(b) shows the $1^{st}$ derivative of the absorbance. Therefore, this figure shows the spectral variation of absorbance and not absorbance.

L. 218: There is an extra "the" to delete.

L. 224: Delete one of the two words "location"

L. 234-235: The RS-LOCAL subset contains 122 soil samples, of which 20 are the local HAFL samples. This subset is also shown in Figure 5. As a result, there is something I do not understand anymore. Figure 2 shows, in case of RS-LOCAL calibration subset, more than 300 samples. Where does this difference come from?

Figure 5. Why, in this case, did the authors prefer to perform PCA analysis on the unscaled and uncentered data?

L. 284: How the color affect the mid-IR spectral absorbance?

L. 300-301: I do not understand this sentence very well, it suggests that mid-IR-FTIR can only be used to analyse transmitted light and not the reflected one, which is not true. There is a need to clarify this.

A general comment on the section 5.1. Although it is interesting to know the different interactions between IR radiations and molecular bonds, I do not see the interest of this section with regard to the aim of this paper.

L. 324-329: Sounds good idea! The authors have, a priori, a dataset that would have allowed them to study the evolution of the RMSE by increments of 10% SC as suggested. Therefore, the question arises as to why this was not done.

---

## Author Comment (AC1)

**Response to reviewers: "Predicting soil carbon by efficiently using variation in a mid-IR soil spectral library" by Helfenstein et al.**

**1 Reviewers**

RC1: https://doi.org/10.5194/soil-2020-93-RC1
5    RC2: https://doi.org/10.5194/soil-2020-93-RC2

**2 Letter of Response**

We thank the reviewers for their comments on our manuscript and their suggestions for improving our work. We have addressed all the comments and as a result our manuscript has improved. Our responses (author comments = AC) are shown in blue text to each of the reviewer's comments (RC) shown in black text.

10   Best regards,
Anatol Helfenstein, on behalf of all authors

**3 Synthesis**

RC1: In this concise paper, the authors have demonstrated that total soil carbon in temperate peatlands, an ecosystem that is underrepresented in the Swiss national database, could be effectively predicted using diffuse reflectance mid-infrared spec-
15   troscopy after adding only a very small number of locally representative samples to the larger database. Studies like this that are demonstrating how to best use national spectral libraries to reduce the cost and effort of collecting new soil information are important next steps in the evolution of soil spectroscopy as a routine tool for soil science. While I think this study is a good contribution to the literature on spectroscopy, I think the authors should do more with the data and analyses as I suggest below. AC: Thank you for this synthesis. We have addressed the comment that more could be done with the data and analyses in the
20   general and specific comments below.

RC2: The topic addressed by the paper is interesting and definitely in the scope of the journal. The article deals with the use of mid-IR spectroscopy in conjunction with chemometrics by means of PLS regression to predict soil carbon content of organic soils collected from two peatland regions of Switzerland. More specifically, the authors propose a statistical and rational approach based on three datasets: i) local dataset from the two peatland regions, ii) national soil spectral library (SSL)
25   spiked with local samples, and iii) subsets containing local and representative SSL samples.
The results of this research work are of interest for the scientific community, particularly those involved in the use of IR spectroscopy for the study and characterization of soils. The proposed approach allows promoting the use of SSL (national or regional) for local predictions while minimizing the number of local reference analyses required. Having said that there are several points that need to be fixed by the authors.
30   AC: Thank you for the synthesis. We have addressed the points you mention in the general and specific comments below.

**4 General comments**

RC1: Why have you chosen to focus on total carbon when the samples have a mix of organic matter and carbonates? There are very few applications of soil information where total carbon is preferred over either total organic carbon or carbonates.
AC: We understand that it may seem counter-intuitive to focus on total carbon (TC) instead of total organic carbon (TOC) and
35   total inorganic carbon (TIC) separately. The practical reason is that we had much more data available on TC in the Swiss SSL. We wanted to test a modelling strategy where we could make use of as much of the SSL information as possible. Furthermore,

using TC instead of e.g. TOC does not influence the overall findings of this study: that a clever combination of extracting relevant observations from a SSL and simple PLSR modelling can greatly reduce field and laboratory work. In other words, this paper is not about modelling different forms of carbon, but rather it demonstrates an efficient spectroscopy modelling approach using large spectral libraries.

RC2: The authors do not satisfactorily explain the relevance of their choice to work on the calibration of total carbon prediction models rather than organic and/or inorganic carbon separately. In most research work, the organic form is given much more attention because of its central role in the functioning of soils.

AC: Please see our response to RC1 above. In addition, we agree with the reviewer that TOC plays a more central role in soil functioning. In our case, given the available data, it made more sense to focus on TC. However, other scientists may also use this spectroscopy modelling approach for TOC.

RC1: This might be splitting hairs but I don't think the three comparisons in this paper should be presented as three different cases. Rather, the local-only models are being compared to two different ways of using the SSL+spike (either as a global PLSR mode or developing an appropriate subset of the SSL using RS-LOCAL before model building). I bring up this comment because of the way the three modeling approaches are first presented in the abstract and intro. A more detailed explanation of the model choices would likely eliminate this comment.

AC: Thank you for raising this point because it is important to us to clearly communicate that this are in fact three very different modelling approaches. The three different approaches are in our opinion clearly explained in the methods section 3.4 "calibration schemes". If we understand your comment correctly, this section made the differences in modelling approaches evident, but you mention that it should be presented more distinctly from the very beginning in the abstract and introduction. We hope that with the addition of a few phrases in the abstract and introduction we can convey that these three approaches are fundamentally different right from the beginning. We now state in the abstract and introduction that the SSL spiking approach uses the entire SSL without any optimization based on the target sites, whereas the RS-LOCAL approach selects useful SSL samples for a particular modelling target.

Perhaps what is confusing is that the jargon "spiking" is defined by its aim to augment an externally defined calibration model with samples from the target site to achieve improved estimation of the considered target variable (Guerrero et al., 2014); but we are aware that there are different ways of achieving this (e.g. Guerrero et al., 2010, 2014; Wetterlind and Stenberg, 2010; Seidel et al., 2019). Here, SSL spiking simply means that representative samples from the target site were selected using the KS algorithm and then added to the entire SSL. This is fundamentally very different from trimming the SSL using RS-LOCAL to find representative samples from the SSL to calibrate an "optimal" model for a new location. The very different results achieved of the SSL spiking vs. RS-LOCAL calibration schemes are also evidence to support these differences in the modelling strategy. Finally, these two approaches are also different from simply using target site samples to train a local model without using a SSL.

RC1: Was there a rationale for not also using a memory-based learning approach? If RS-LOCAL can achieve results just as good as or better than MBL, then that is a good argument for the simplicity of RS-LOCAL. Given the diversity in SC content, carbonate content, and soil type in the HAFL dataset, it seems that one subset of the SSL is not going to be as good as subsets specifically built for the individual samples.

AC: To reply to this suggestion, we would like to refer to Lobsey et al. (2017), where RS-LOCAL is first presented and explained in full detail. In this study, the authors compare memory-based learning (MBL) and other methods to RS-LOCAL and showed that RS-LOCAL performed better. In fact, the main reason to develop RS-LOCAL was to overcome the limitations of methods such as MBL. While MBL can outperform general modelling approaches not targeted towards a new location, there is evidence to suggest that it will perform worse for case studies such as ours, where new samples from a target area are not covered well in the variability of the SSL. Memory-based methods such as different variants of the spectrum-based learner are founded on the principle of spectral relatedness and chemical relatedness being reflected by distances between objects in the library and the multivariate space. This optimization can fail when data from the library that is considered as similar is too far to be chemically related or when close-by samples have inadequate coverage of the response or density distributions of the spectral variables to accurately model soil properties of the unknown observations in the data set. The PCA of the SSL and target data (HAFL)

show that many of the target samples only overlap with a few SSL samples (Figure 5), suggesting that for many of the HAFL samples, it is unlikely that MBL can find good "subsets specifically built for the individual samples", as you mention in your comment.

RC1: The size of the validation sets are changing with increasing number of spike samples. This is setting up a situation where the results are not perfectly comparable in Fig 8. How about restricting your validation set to the 58 samples that are never used in calibration?

AC: Thank you for this comment. We indeed realized that model validations are not 100% comparable if we do not always use the same exact samples. We agree with the reviewer and have now changed our results so that no matter the number of samples used in model calibration in the three compared approaches, the models are validated always using the same exact 58 samples.

RC2: The approach for the calibration of local models, by considering calibration set with data (15, 20, 25, 30 . . . ) less or equal to that of the validation set, is statistically questionable. Moreover, how robust are these calibration models? What would happen if we made 5 random draws instead of the KS algorithm? The classical approach consists in in splitting a data set, taking 2/3 for calibration and 1/3 for validation. So why the authors did not go so far as to test this classical subdivision?

AC: We strongly disagree with the reviewer's implications here. We have shown in this study that as few as 5-10 target site soil samples are required to calibrate accurate models when also using SSL information, as indicated by the validation metrics of the remaining samples. We do not understand why this would be "statistically questionable" only because we are not following other modelling approaches that aim for a completely different goal. The independent validation plots and metrics using samples not used in model calibration show how robust the models are. We show that local and RS-LOCAL models are more robust than simply adding local samples to the entire SSL and that using RS-LOCAL has a further advantage over local models because less new samples are required.

The "classical subdivision" with 2/3 for calibration and 1/3 for validation does not make sense for the aim of this study because the efficiency would be decreased, i.e. it would require more laborious field and laboratory work. Besides, un-favourable bias-variance trade-off of this single hold-out method for calibrating and validating models compared to $K$-fold cross-validation (reducing variance of performance estimators via multiple splits) has been repeatedly shown by fundamental studies in statistical learning.

The Kennard-Stone algorithm selects by default first the calibration samples, that uniformly covers the feature space of the spectra. Since the points that are farthest apart in the Mahalanobis space are selected fist, they lay at the edge of the multivariate cloud and tend to include extreme points. This can lead to over-optimistic performance assessments since atypical data points are excluded from validation. We achieve careful and rather conservative validations of our modeling scenarios because we purposely inverted this classical sequence of assignment, which meant that validation samples that include boundary points were selected first.

RC1: I'm struggling with the high bias in the SSL validation results. Could this be due to a limit of 10 components for the PLSR model? With 4000+ samples, it could well be justified to search up to 20 or even 30 components to find the minimum in RMSE. I'm also wondering if you transformed the TC data prior to model fitting (log or square root transform) if that would get rid of some of the curvilinear nature to the fit – Baldock et al. (2013 Soil Research - https://doi.org/10.1071/SR13077) found that sq. root transformation really improved model fits. It is very much possible that the bias is real but it would be good to test these two ideas out to see if it helps remove the bias.

AC: We disagree with the suggestions of this comment. We think it is logical that calibrating a model where the vast majority of SSL samples are mineral soils with carbon amounts between $0\,\%$ to $5\,\%$ TC to predict samples with a uniform continuous distribution between $0\,\%$ to $52\,\%$ TC will introduce a large bias. Table 1 shows how different the TC distributions of the SSL are compared to the target HAFL data, so it is by no means a surprise that calibrating a statistical model will yield bad predictions for a new location with a completely different distribution.

We have tested the results when increasing the number of components to 15 components but there was no decrease in bias. Moreover, we do not think that simply increasing the number of components or transforming the data will improve results, nor does it serve the purpose of this study. It is evident that without selecting representative samples from the SSL, the predictive

performance of new and very different samples will remain poor. Regarding the transformation suggestion, this may improve
130 the model fit of the transformed response, but the goal is to produce accurate predictions using new data and as soon as back-transforming the predictions it will be evident that predictive performance is still low. This highlights one of the main findings of this study: using a SSL is only effective for dissimilar target sites if incorporating a method to mine and reduce the SSL effectively, exemplified here with the use of RS-LOCAL.

RC1: I'm not sure section 5.1 is necessary or even really fair to include in this paper. The topic is certainly interesting – can
135 MIR be used to study peat composition? As discussed there is a growing literature doing just this. Where I see a problem is that most of those papers focused exclusively on pure peat soils while the HAFL set of samples compose everything from pure OM to almost pure mineral soil. As your Fig 4 shows, the majority of the variability in the HAFL set comes from this gradient in C content.
AC: Thank you for addressing this point and we now also see that the second paragraph of section 5.1 about peat composition
140 is not relevant for this paper. We neither have the data nor conducted the analyses to discuss this topic. In addition to focusing on TOC instead of TC, we would need to have conventional reference measurements of additional peat properties such as functional group compositions, which is not the focus of this study and outside its scope.

RC2: The authors used the following statistical criteria R², bias and RMSE to evaluate the prediction performance of spectral models in both calibration and validation. Most of the scientist involved in the field of soil spectroscopy also use RPD or even
145 RPIQ (recommended when the distribution is not normal) criterion. I think the authors should provide them so that they are available to readers who would like to compare their results.
AC: Thank you for this comment and we agree that these additional metrics might be useful to compare our study to others. We have included them in the revised version.

**5 Specific comments**

150 RC1: Title – Given the focus and brevity of this MS I think a more focused title on applying a national library to peatlands is appropriate. The phrase "efficiently using variation" will not mean much to most readers of Soil.
AC: Thank you for addressing this. We have changed the title to "Quantifying soil carbon in temperate peatlands using a mid-IR soil spectral library".

RC2: Title: Maybe it should refer to peatlands to make it a bit more explicit.
155 AC: Please see above comment.

**5.1 Abstract**

RC1: L8 What do you mean by "organo-mineral diversity"?
AC: We have now deleted the phrase "and their organo-mineral diversity".

RC1: L11 "target-feature representations" is jargon that should be avoided in the abstract. Can this be reworded?
160 AC: This sentence has been rewritten to "The focus was on minimizing the need for new reference analyses by efficiently mining the spectral information of the SSL."

RC1: L15-19 I found this summary of the results really difficult to digest until I had read the entire paper. Please rephrase and simplify.
AC: The last sentence in these lines has been simplified to "However, calibrations using RS-LOCAL only required 5 local
165 samples for very accurate models (RMSE = 2.9 %), while purely local calibrations required 50 samples for similarly accurate results (RMSE < 3 %).

RC1: L23 "a SSL" – do you mean any SSL or the Swiss SSL here?
AC: We changed it to "the SSL", implying the Swiss SSL. We cannot guarantee with 100 % certainty that any SSL is always sufficient, e.g. if it is very small or only covers a small part of the total variability of soils in a region or country.

**5.2 Introduction**

RC1: L63-64 This sentence was difficult to understand intent. Please rephrase.
AC: We have rephrased this to "On the other hand, the focus of these studies may be explained by the fact that in the past, large-scale models still tended to perform less accurate than small-scale models."

RC1: L78-82 Can you please spend some time defining in simple terms what memory based learning and transfer learning mean? Few soil scientists will be familiar with these terms.
AC: We have defined transfer learning on L82. However, to make it more clear and also define memory- or instance-based learning, we have re-written L78-83 as follows:
   "These methods can be summarized as spiking (Shepherd and Walsh, 2002; Brown, 2007; Wetterlind and Stenberg, 2010; Seidel et al., 2019), subsetting (Araújo et al., 2014; Lobsey et al., 2017), memory- or instance-based learning (Ramirez-Lopez et al., 2013; Gholizadeh et al., 2016), or transfer learning (Padarian et al., 2019). Spiking can be defined as adding local soil samples to a general SSL. Subsetting can generally be defined as diving the SSL into smaller partitions based on characteristic features (e.g. geographic regions, soil type, etc.) or a specific method. One such method is memory- or instance-based learning, in which soil samples similar or related to the target local samples are retrieved from memory and merged to calibrate a new model. Finally, transfer learning is the process of sharing intra-domain information and rules learnt by general models to a local domain (Pan and Yang, 2010)."

RC1: L110 Change to modelling approaches "for" SC
AC: This was changed as suggested.

**5.3 Soil Data: The Swiss SSL and the HAFL Dataset**

RC1: L115 Shouldn't this be in the methods section?
AC: The reason for separating this was that in this study, we did not measure and perform all the analyses of the entire SSL, merely the HAFL samples. Hence, we did not want to present the methods involved to establish the SSL as our own, i.e. it was already there. But it might make more sense to add it to the methods sections anyway. We have moved it and it is now the first subsection of the methods section.

RC2: L. 116-119: The BDM database is made up of soil samples collected from topsoil (0-20 cm), is it the same for NABO samples?
AC: Yes, it is the same for the NABO samples. We changed this sentence to "The current Swiss SSL in the mid-IR range consists of 3723 topsoil ($0\,cm$ to $20\,cm$) samples from 1094 locations from the Biodiversity Monitoring program (BDM; e.g. FOEN, 2018; Meuli et al., 2017) and 572 topsoil samples from 71 locations from the National Soil Monitoring Network (NABO; Table 1 and Figure 1; e.g. NABO, 2018; Gubler et al., 2015).

RC2: L. 120: The organic soils are not sufficiently represented in the SSL. It would nevertheless be interesting to know their amount (or fraction) and their range of variation in SC content.
AC: This is a good point. We have now included what proportion of the SSL and HAFL samples can be considered organic soils according to WRB and the range of variation among them.

RC2: L. 122-123: the local HAFL consists of samples collected from 0 up to 2 m depth. I assume these are soil cores that were collected. In that case, what was the thickness of the soils taken for the reference analyses and what were the depths considered?
AC: The samples are partly from soil profiles and partly from manual drills. On each location samples from three to seven soil horizons were taken. These horizons have thicknesses from 5 cm to several dm. A sample from the topsoil (the first or second horizon) is always included. The sites are often characterized by a formerly dynamic change between alluvial stages and peat accumulation stages. Therefore, the variation of SOC within a site is enormous. We tried to reproduce this variance by taking samples from all characterizing horizons.

RC1: L125-127 Please rephrase this sentence, it is currently difficult to read.

AC: We have rephrased this sentence to: "These samples originate from either undisturbed and waterlogged organic horizons, mineralized organic horizons under agricultural use, horizons with sandy or calcareous substrate material or horizons containing a mixture of these characteristics."

**5.4 Methods**

RC1: L130 What method was used for grinding? This would be nice to include given some of the current debate of grinding.

AC: We used the ball-mill method. We will add this to the methods.

RC1: L130 What does it mean to optimize signal-to-noise? Wouldn't we want to maximize S:N?

AC: We have changed "optimize" to "maximize".

RC1: L138-146 I'd prefer to see preprocessing discussed with the rest of the data analysis

AC: This section has been changed so that we mention the different preprocessing steps in the methods section but explain the final preprocessing steps we ended up using in the results section with the rest of the data analysis.

RC1: L138 If you tested several pre-processing approaches, please show the results of this testing as supplemental information. It would be interesting to know how much better the SG 1st derivative performed relative to just baseline correction.

AC: What does the reviewer mean by "baseline correction". Among the preprocessing steps we tested, we did not test for a case without using the SG filter at all. However, we have attached the difference in results for selecting every 8th variable as opposed to every variable in the model building. This was the main preprocessing step and is directly related to the window size (e.g. smaller window relates to more variables).

RC1: L150 What was the TC method for NABO samples?

AC: TC was measured the same way for both NABO and BDM samples by dry combustion using the CHN628 Series Elemental Determinator from LECO. We have now changed this so that we explicitly state "...between the two different CHN628 Series Elemental Determinators used (NABO and BDM samples used the same machine)..." [L153].

RC1: L163 Was tuning based on RMSE or R2?

AC: Tuning was based on RMSE. We have now clearly stated this.

RC1: L167 Why did you limit PLSR to only 10 components especially when using the entire SSL? This might be a reason for the high bias in the results.

AC: We disagree with the suggestions of this comment. We think it is logical that calibrating a model where the vast majority of SSL samples are mineral soils with carbon amounts between $0\%$ to $5\%$ TC to predict samples with a uniform continuous distribution between $0\%$ to $52\%$ TC will introduce a large bias. Table 1 shows how different the TC distributions of the SSL are compared to the target HAFL data, so it is by no means a surprise that calibrating a statistical model will yield bad predictions for a new location with a completely different distribution.

We have tested the results when increasing the number of components to 15 components but there was no decrease in bias. Moreover, we do not think that simply increasing the number of components will improve the results, nor does it serve the purpose of this study. It is evident that without selecting representative samples from the SSL, the predictive performance of new and very different samples will remain poor. This highlights one of the main findings of this study: using a SSL is only effective for dissimilar target sites if incorporating a method to mine and reduce the SSL effectively, exemplified here with the use of RS-LOCAL.

RC1: L168 If you ran cubist and don't show the results, then why even mention it here? Given this is a really short paper, I'd like to see the Cubist results included.

AC: This paper is not about comparing different general modelling approaches, e.g. PLSR vs. Cubist. Rather, it is about the comparison of a general modelling approach to a flexible approach that includes data-mining a SSL for local applications and

how this can improve prediction accuracy and efficiency. We think it has been repeatedly shown that Cubist and PLSR both work well for spectroscopic modelling and that as a scientific community, we should go a step further than simply comparing one general model to another. Using RS-LOCAL in combination with an SSL can just as easily be done using Cubist instead of PLSR to calibrate a model. We do not show it here for simplicity so as to not deviate from the story-line.

RC2: L. 168-169: No need to talk about Cubist models since the results are not included in this article.
AC: We agree with this comment and have now removed this information that we also tested Cubist.

RC1: L180 Why did you choose to stop at a 50/50 cal/val split? Many studies often use 70/30 and 80/20 splits?
AC: We strongly disagree with the reviewer's implications here. We have shown in this study that as few as 5-10 target site soil samples are required to calibrate accurate models when also using SSL information, as indicated by the validation metrics of the remaining samples. The independent validation plots and metrics using samples not used in model calibration show how robust the models are. We show that local and RS-LOCAL models are more robust than simply adding local samples to the entire SSL and that using RS-LOCAL has a further advantage over local models because less new samples are required. The "classical subdivision" with 2/3 for calibration and 1/3 for validation does not make sense for the aim of this study because the efficiency would be decreased, i.e. it would require more laborious field and laboratory work. Besides, unfavourable bias-variance trade-off of this single hold-out method for calibrating and validating models compared to $K$-fold cross-validation (reducing variance of performance estimators via multiple splits) has been repeatedly shown by work in statistical learning.

RC1: L180 Why was KS used for sample selection and not conditioned Latin hypercube sampling or other techniques? I don't think there is anything wrong with KS but some justification would be good.
AC: We used KS as opposed to other techniques simply because it worked well in two case studies in Australia and New Zealand and for the development of RS-LOCAL (Lobsey et al., 2017), as well as in this study. The main message of this paper is that efficiently using a SSL can help to reduce time consuming field and laboratory work in quantifying soil carbon, even if the soils are very different from most of the SSL soils. This will be the case no matter the selection method chosen. Applications of this study may also use cLHS or fuzzy c-means (FCMS) sampling if this is preferred. The optimal selection method will also be a function of the structure and variability in the dataset and different for each application.
   The Kennard-Stone algorithm selects by default first the calibration samples, that uniformly covers the feature space of the spectra. Since the points that are farthest apart in the Mahalanobis space are selected fist, they lay at the edge of the multivariate cloud and tend to include extreme points. This can lead to over-optimistic performance assessments since atypical data points are excluded from validation. We achieve careful and rather conservative validations of our modeling scenarios because we purposely inverted this classical sequence of assignment, which meant that validation samples that include boundary points were selected first.

RC1: L181-183 Are these sentences included to justify not having locals models of less than 15 samples as done with the SSL and RS-LOCAL? If yes, it may be easier just to state that you did not build models of less than 15 samples.
AC: Indeed they were only included due to this. We have now removed this and simply state that we did not build models of less than 15 samples.

RC1: L188 For the SSL and RS-LOCAL models, why didn't you try a no-spike case study? It would be really interesting to know how the SSL stands up with no new local information.
AC: We specifically lowered the number of local samples until we could see the expected clear jump in model performance (Figure 8). In this case this big jump was between using 3 and 5 samples, with very poor model performance when using only 3 local samples and starting at 5 local samples remaining relatively steady for the RS-LOCAL approach. Since model accuracy is already so low when using only 3 local samples, we do not see the need to further lower the number local samples to 0.

RC1: L195 Can you better explain how RS-LOCAL searches for subset K and how you ended up with K being the same for all levels of m?

295   AC: The details of how RS-LOCAL searches for subset $K$ are explained in seven chronological steps in Lobsey et al. (2017), which we reference in this manuscript. We have added a sentence explaining that $K$ remains the same for all values of m only as long as $k$, $b$, $r$ and the size of the entire SSL remain constant.

RC1: L212 A short section describing how you evaluated the results of the different models would be helpful.
AC: We clearly state in L170-172 that RMSE, bias, $SS_{mse}/R^2$ were used to assess the error (accuracy) and model fit. We also
300   state that each calibrated model iteration was evaluated using the validation set (e.g. L185-186, L193-194, L211-212).

RC1: Fig 2 I found this figure to be very helpful, thanks for including. I suggest either changing the color or using hash marks for NABO as you cannot see that part of the bar in black and white.
AC: Thank you for this suggestion. We have changed the plot so that the distinct datasets can also be seen in black and white.

**5.5   Results**

305   RC1: L218 Picking only one region (2000 cm-1) to highlight here seems arbitrary. There are other regions with clear patterns with SC.
AC: This is true. We have changed this so that we highlight several regions with clear patterns for SC.

RC2: L. 218-219: There is an ambiguity in this sentence: "where soils with a low carbon content had higher absorbance than soils with a high carbon content". According to the pre-processing carried out by the authors, Figure 3(b) shows the 1st
310   derivative of the absorbance. Therefore, this figure shows the spectral variation of absorbance and not absorbance.
AC: Thank you for catching this. We have changed the sentence to "Pre-processed absorbance values showed a clear pattern at specific wavenumbers according to the SC content, such as around 1900 cm$^{-1}$, 2000 cm$^{-1}$, 2900 cm$^{-1}$, or 3600 cm$^{-1}$."

RC2: L. 218: There is an extra "the" to delete.
AC: Thank you for catching this typo. The first "the" was replaced by a "to".

315   RC1: Fig 3 & 4 The color ramp is not colorblind friendly and does not reproduce in black and white. There are lots of great colorblind friendly options available at https://colorbrewer2.org/ and other websites.
AC: Thanks for the suggestion. We have adjusted to a color-blind and black and white friendly color ramp (viridis, option "magama").

RC2: L. 224: Delete one of the two words "location"
320   AC: We deleted one the duplicate "location".

RC1: L229 Can you add a measure of skewedness to Table 1?
AC: We have now added a measure of skewedness to Table 1, as suggested.

RC1: L231 Please tell us how many samples in the Swiss SSL are organic soils. To me there seems to be fairly good coverage of the HAFL data.
325   AC: We have now included in the dataset description the number of samples considererd organic soils in the SSL according to WRB (IUSS Working Group WRB, 2014).

RC1: L233 There are quantitative ways of assessing differences in PC space such as calculating centroids and hulls. You can also calculate a resemble matrix and then apply multivariate ANOVA analysis to be truly quantitative here.
AC: Thank you for these suggestions. We have calculated the centroids and hulls specific to each dataset for additional com-
330   parison and will split figure 5 into the scatterplot of the first two PCs (as it currently is) and a plot of centroid and hulls of each dataset. We split these in order to prevent a overly noisy plot with too much information that it becomes difficult to read. We will also calculate a resemble matrix and apply a multivariate ANOVA.

RC2: L. 234-235: The RS-LOCAL subset contains 122 soil samples, of which 20 are the local HAFL samples. This subset is also shown in Figure 5. As a result, there is something I do not understand anymore. Figure 2 shows, in case of RS-LOCAL calibration subset, more than 300 samples. Where does this difference come from?

AC: Thank you for catching this! This is a mistake because in figure 2, it shows RS-LOCAL calibration schemes were $k = 300$. However, in all the other figures, we show results for $k = 100$ because this showed slightly better results. To avoid confusion, we will therefore change figure 2 so that the calibration scheme shows the number of RS-LOCAL samples for $k = 100$ just as in the other figures.

RC1: Fig 5 I suggest using very light symbols for BDM samples not used in validation. Also move these symbols to the back; they currently cover most of the other data making this figure difficult to read.

AC: We want to keep the same color scheme of the three datasets as in the rest of the figures. However, as you suggest, we will move the BDM symbols to the back and try to use lighter symbols to improve readability.

RC2: Figure 5. Why, in this case, did the authors prefer to perform PCA analysis on the unscaled and uncentered data?

AC: We explain in the caption of figure 5 that we used an unscaled and uncentered PCA "since it better showed the data distribution than a scaled and centered PCA."

RC1: L240 I'd like to see the RS-LOCAL tuning results presented in the supplement. I'm curious as to how important each parameter was.

AC: We will add plots of $R^2$, bias and RMSE for setting different values of $k$ in RS-LOCAL of both calibration and more importantly, validation results to the supplements. The other parameters $b$ and $r$ made little difference as long as the suggestions of Lobsey et al. (2017) were followed.

RC1: Fig 6 & 7 are not very informative given how you are trying to discuss results across increasing number of spike samples. Fig 8 is really the only figure that really matters for interpreting the results presented in section 4.3. I would suggest either moving both figures to supplement or just keep Fig 6 as an example of the different model fits at one level of spiking.

AC: We agree with this comment and will move figure 7 to the supplements. As you indicate, we think it is helpful for to keep figure 6 as one example of predicted vs. observed of model calibration and validation of the different modelling approaches.

RC1: Fig 8 Bias is also in units of %. You might want to mention in the caption that RMSE axis is log.

AC: We will add the units in % to the bias y-axis label. Please note that we already mention that RMSE is in log scale in the caption of figure 8.

**5.6 Discussion**

RC2: A general comment on the section 5.1. Although it is interesting to know the different interactions between IR radiations and molecular bonds, I do not see the interest of this section with regard to the aim of this paper.

AC: We agree that the second paragraph of section 5.1 about peat composition is not relevant for this paper. We neither have the data nor conducted the analyses to discuss this topic. In addition to focusing on TOC instead of TC, we would need to have conventional reference measurements of additional peat properties such as functional group compositions, which is not the focus of this study and outside its scope. We will therefore remove the second paragraph.

However, we would like to keep the first paragraph of section 5.1 explaining some interactions between mid-IR and different TC constituents. Although it is not the main focus of this paper, it can help to explain why there are such clear patterns between TC and the preprocessed spectra (Figure 3) as well as the PCs (Figure 4). This also serves as a link to more chemometric-focused studies.

RC1: L282-283 Are you referring to overall correlation or to specific bands with this sentence? It is unclear.

AC: This refers to the overall correlation of the spectra across (almost) all wavenumbers and at the same time acts as a topic sentence to introduce the topic of discussion in this paragraph. We have added "overall" so that the new sentence is more clear:

"The measured and pre-processed mid-IR soil spectra and PCA results of all local HAFL samples revealed high correlation
375     between the spectral absorbance values and a broad range of SC content (Figures 3 & 4)."

RC2: L. 284: How the color affect the mid-IR spectral absorbance?
AC: We have removed color from this sentence since it should not have an influence in the mid-IR range. The sentence now reads "According to past studies that assessed variable importance, we assumed that soil texture and mineralogical and organic composition influence mid-IR spectral absorbance the most (Madari et al., 2006; Bornemann et al., 2010; Calderón et al.,
380     2011)."

RC1: L300 Are you saying these are transmission measurements instead of diffuse reflectance? I'm not familiar with all of these studies but I know the Matamala et al 2019 study used diffuse reflectance. A really interesting study using MIR spectroscopy on peat soils to look at decomposition state by correlating MIR data with NMR data was done by Hodgkins et al (2018 Nature Communications - https://www.nature.com/articles/s41467-018-06050-2).
385     AC: Thank you for noting this and you are correct that Matamala et al. (2019) indeed used DRIFT mid-IR spectroscopy. We have removed this paragraph from the discussion entirely so this no longer applies. But thank your also for recommending the Hodgkins et al. (2018) paper; I have added it to my library.

RC2: L. 300-301: I do not understand this sentence very well, it suggests that mid-IR-FTIR can only be used to analyse transmitted light and not the reflected one, which is not true. There is a need to clarify this.
390     AC: This was indeed unclear. This is no longer necessary since we have removed this paragraph from the discussion entirely.

RC1: L309 Do you think the strong bias in the spiked SSL models is a common feature or is it specific to your application of an SSL to peatlands? Or is it an artifact of using PLSR instead of Cubist or other ML models.
AC: In general, we think that a strong bias when using the entire SSL to predict new samples is a common feature. In our case, we expected this bias to be particularly high, since, as you mention, we are using it to target peatlands specifically, which
395     are only marginally represented in the current Swiss SSL. We know from our results using Cubist that this did not change the strong bias. It is possible that more complex models such as neural networks may perform slightly better here, but this is not the point. Instead of simply changing the general model, we suggest to build target-specific models by adequately mining a SSL. This use of transfer learning yields a much bigger improvement in most if not all cases than a general modelling approach using all the SSL samples. There may however, be exceptions to this strong bias if the target dataset has a very similar distribution
400     of soil property values as in the SSL. But ideally, we would like to be able to use the SSL for all kinds of situations including the "extremes".

RC1: L313 You say that you cannot infer if lower SC range is better predicted than the higher SC range but you have the data to do exactly such a test. You certainly have the space to divide out the validation results into low v. high C ranges to better see where the bias and lack of fit is creeping in.
405     AC: Thanks for your suggestion. In addition to the current results of model accuracy over the entire scale of TC, we will add the model performance measures across increments of TC to see if model accuracy decreases with increasing values of the response.

RC2: L. 324-329: Sounds good idea! The authors have, a priori, a dataset that would have allowed them to study the evolution of the RMSE by increments of 10% SC as suggested. Therefore, the question arises as to why this was not done.
410     AC: Thanks for your suggestion. Please see our comment above. We will test this in the revised version.

RC1: L325-326 This sentence makes it sound like you did exactly what I just asked for in the last comment but did not present the findings. This sentence cannot be in the discussion if you do not present the data elsewhere.
AC: We understand there is some confusion here. The overall modelling results are already presented in figure 8 for the case where the number of local HAFL samples (x-axis) equals 3. We will now reference figure 8 in parentheses at the end of this
415     sentence so that it is more clear. In addition, we will test the results on model validation accuracy as you suggest above.

RC1: L330-331 In general be careful about making statements about being the first to do something. There are very few truly original ideas in science. In this case, the USDA NRCS KSSL MIR spectral library contains 2000 organic soil samples – see papers by Wijewardane et al. 2018 and Dangal et al. 2019.

AC: We fully agree with the reviewer and we will make sure to be careful about such claims. We have reformulated this sentence and added the references to the papers you mention.

RC1: L346 – Ramirez-Lopez et al (2019 EJSS - https://doi.org/10.1111/ejss.12752) incorporated FTIR prediction uncertainty into digital soil mapping.

AC: Thank you for this suggestion. We now reference Ramirez-Lopez et al 2019 as an example of a farm-scale digital soil mapping application using soil spectroscopy as point data and how uncertainty propagates into the soil maps. In addition to the paper you suggest, we have now done a more thorough search of error propagation analysis of spectroscopic models and also reference Brodský et al. (2013) and Viscarra Rossel et al. (2016).

**References**

Araújo, S. R., Wetterlind, J., Demattê, J. a. M., and Stenberg, B.: Improving the Prediction Performance of a Large Tropical Vis-NIR Spectroscopic Soil Library from Brazil by Clustering into Smaller Subsets or Use of Data Mining Calibration Techniques, European Journal of Soil Science, 65, 718–729, https://doi.org/10.1111/ejss.12165, 2014.

Bornemann, L., Welp, G., and Amelung, W.: Particulate Organic Matter at the Field Scale: Rapid Acquisition Using Mid-Infrared Spectroscopy, Soil Science Society of America Journal, 74, 1147–1156, https://doi.org/10.2136/sssaj2009.0195, 2010.

Brodský, L., Vašát, R., Klement, A., Zádorová, T., and Jakšík, O.: Uncertainty Propagation in VNIR Reflectance Spectroscopy Soil Organic Carbon Mapping, Geoderma, 199, 54–63, https://doi.org/10.1016/j.geoderma.2012.11.006, 2013.

Brown, D. J.: Using a Global VNIR Soil-Spectral Library for Local Soil Characterization and Landscape Modeling in a 2nd-Order Uganda Watershed, Geoderma, 140, 444–453, https://doi.org/10.1016/j.geoderma.2007.04.021, 2007.

Calderón, F. J., Reeves, J. B., Collins, H. P., and Paul, E. A.: Chemical Differences in Soil Organic Matter Fractions Determined by Diffuse-Reflectance Mid-Infrared Spectroscopy, Soil Science Society of America Journal, 75, 568–579, https://doi.org/10.2136/sssaj2009.0375, 2011.

FOEN: Biodiversity Monitoring Switzerland (BDM), http://www.biodiversitymonitoring.ch/ en/home.html, 2018.

Gholizadeh, A., Borůvka, L., Saberioon, M., and Vašát, R.: A Memory-Based Learning Approach as Compared to Other Data Mining Algorithms for the Prediction of Soil Texture Using Diffuse Reflectance Spectra, Remote Sensing, 8, 341, https://doi.org/10.3390/rs8040341, 2016.

Gubler, A., Peter, S., Wächter, D., Meuli, R., and Keller, A.: Ergebnisse der Nationalen Bodenbeobachtung (NABO) 1985-2009. Zustand und Veränderungen der anorganischen Schadstoffe und Bodenbegleitparameter., Tech. rep., Bundesamt für Umwelt (BAFU), Bern, 2015.

Guerrero, C., Zornoza, R., Gómez, I., and Mataix-Beneyto, J.: Spiking of NIR Regional Models Using Samples from Target Sites: Effect of Model Size on Prediction Accuracy, Geoderma, 158, 66–77, https://doi.org/10.1016/j.geoderma.2009.12.021, 2010.

Guerrero, C., Stenberg, B., Wetterlind, J., Rossel, R. A. V., Maestre, F. T., Mouazen, A. M., Zornoza, R., Ruiz-Sinoga, J. D., and Kuang, B.: Assessment of Soil Organic Carbon at Local Scale with Spiked NIR Calibrations: Effects of Selection and Extra-Weighting on the Spiking Subset, European Journal of Soil Science, 65, 248–263, https://doi.org/10.1111/ejss.12129, 2014.

Hodgkins, S. B., Richardson, C. J., Dommain, R., Wang, H., Glaser, P. H., Verbeke, B., Winkler, B. R., Cobb, A. R., Rich, V. I., Missilmani, M., Flanagan, N., Ho, M., Hoyt, A. M., Harvey, C. F., Vining, S. R., Hough, M. A., Moore, T. R., Richard, P. J. H., De La Cruz, F. B., Toufaily, J., Hamdan, R., Cooper, W. T., and Chanton, J. P.: Tropical Peatland Carbon Storage Linked to Global Latitudinal Trends in Peat Recalcitrance, Nature Communications, 9, 3640, https://doi.org/10.1038/s41467-018-06050-2, 2018.

IUSS Working Group WRB: World Reference Base for Soil Resources 2014. International Soil Classification System for Naming Soils and Creating Legends for Soil Maps., World Soil Resources Reports No. 106, FAO, Rome, 2014.

Lobsey, C. R., Viscarra Rossel, R. A., Roudier, P., and Hedley, C. B.: Rs-Local Data-Mines Information from Spectral Libraries to Improve Local Calibrations, European Journal of Soil Science, 68, 840–852, https://doi.org/10.1111/ejss.12490, 2017.

Madari, B. E., Reeves, J. B., Machado, P. L., Guimarães, C. M., Torres, E., and McCarty, G. W.: Mid- and near-Infrared Spectroscopic Assessment of Soil Compositional Parameters and Structural Indices in Two Ferralsols, Geoderma, 136, 245–259, https://doi.org/10.1016/j.geoderma.2006.03.026, 2006.

Matamala, R., Jastrow, J. D., Calderón, F. J., Liang, C., Fan, Z., Michaelson, G. J., and Ping, C.-L.: Predicting the Decomposability of Arctic Tundra Soil Organic Matter with Mid Infrared Spectroscopy, Soil Biology and Biochemistry, 129, 1–12, https://doi.org/10.1016/j.soilbio.2018.10.014, 2019.

Meuli, R. G., Wächter, D., Schwab, P., Kohli, L., and Zimmermann, R.: Connecting Biodiversity Monitoring with Soil Inventory Data – A Swiss Case Study, 2017.

NABO: Swiss Soil Monitoring Network (NABO), https://www.agroscope.admin.ch/agroscope/ en/home/themen/umwelt-ressourcen/boden-gewaesser-naehrstoffe/nabo.html, 2018.

Padarian, J., Minasny, B., and McBratney, A. B.: Transfer Learning to Localise a Continental Soil Vis-NIR Calibration Model, Geoderma, 340, 279–288, https://doi.org/10.1016/j.geoderma.2019.01.009, 2019.

Pan, S. J. and Yang, Q.: A Survey on Transfer Learning, IEEE Trans. Knowl. Data Eng., 22, 1345–1359, https://doi.org/10.1109/TKDE.2009.191, 2010.

Ramirez-Lopez, L., Behrens, T., Schmidt, K., Stevens, A., Demattê, J. A. M., and Scholten, T.: The Spectrum-Based Learner: A New Local Approach for Modeling Soil Vis–NIR Spectra of Complex Datasets, Geoderma, 195-196, 268–279, https://doi.org/10.1016/j.geoderma.2012.12.014, 2013.

Seidel, M., Hutengs, C., Ludwig, B., Thiele-Bruhn, S., and Vohland, M.: Strategies for the Efficient Estimation of Soil Organic Carbon at the Field Scale with Vis-NIR Spectroscopy: Spectral Libraries and Spiking vs. Local Calibrations, Geoderma, 354, 113 856, https://doi.org/10.1016/j.geoderma.2019.07.014, 2019.

Shepherd, K. D. and Walsh, M. G.: Development of Reflectance Spectral Libraries for Characterization of Soil Properties, Soil Science
480    Society of America Journal, 66, 988–998, https://doi.org/10.2136/sssaj2002.9880, 2002.

Viscarra Rossel, R., Brus, D., Lobsey, C., Shi, Z., and McLachlan, G.: Baseline Estimates of Soil Organic Carbon
by Proximal Sensing: Comparing Design-Based, Model-Assisted and Model-Based Inference, Geoderma, 265, 152–163,
https://doi.org/10.1016/j.geoderma.2015.11.016, 2016.

Wetterlind, J. and Stenberg, B.: Near-Infrared Spectroscopy for within-Field Soil Characterization: Small Local Calibrations Compared
485    with National Libraries Spiked with Local Samples, European Journal of Soil Science, 61, 823–843, https://doi.org/10.1111/j.1365-
2389.2010.01283.x, 2010.